# Causal Reinforcement Learning: A Survey

**Zhihong Deng**                                                                 *zhi-hong.deng@student.uts.edu.au*
*Australian Artificial Intelligence Institute, University of Technology Sydney*

**Jing Jiang**                                                                              *jing.jiang@uts.edu.au*
*Australian Artificial Intelligence Institute, University of Technology Sydney*

**Guodong Long**                                                                      *guodong.long@uts.edu.auu*
*Australian Artificial Intelligence Institute, University of Technology Sydney*

**Chengqi Zhang**                                                                    *chengqi.zhang@uts.edu.au*
*Australian Artificial Intelligence Institute, University of Technology Sydney*

**Reviewed on OpenReview:** *https://openreview.net/forum?id=qqnttX9LPo*

## Abstract

Reinforcement learning is an essential paradigm for solving sequential decision problems under uncertainty. Despite many remarkable achievements in recent decades, applying reinforcement learning methods in the real world remains challenging. One of the main obstacles is that reinforcement learning agents lack a fundamental understanding of the world and must therefore learn from scratch through numerous trial-and-error interactions. They may also face challenges in providing explanations for their decisions and generalizing the acquired knowledge. Causality, however, offers notable advantages by formalizing knowledge in a systematic manner and harnessing invariance for effective knowledge transfer. This has led to the emergence of causal reinforcement learning, a subfield of reinforcement learning that seeks to enhance existing algorithms by incorporating causal relationships into the learning process. In this survey, we provide a comprehensive review of the literature in this domain. We begin by introducing basic concepts in causality and reinforcement learning, and then explain how causality can help address key challenges faced by traditional reinforcement learning. We categorize and systematically evaluate existing causal reinforcement learning approaches, with a focus on their ability to enhance sample efficiency, advance generalizability, facilitate knowledge transfer, mitigate spurious correlations, and promote explainability, fairness, and safety. Lastly, we outline the limitations of current research and shed light on future directions in this rapidly evolving field.

## 1 Introduction

> *"All reasonings concerning matter of fact seem to be founded on the relation of cause and effect. By means of that relation alone we can go beyond the evidence of our memory and senses."*
>
> *—David Hume, An Enquiry Concerning Human Understanding.*

Humans possess an inherent capacity to grasp the concept of causality from a young age (Wellman, 1992; Inagaki & Hatano, 1993; Koslowski & Masnick, 2002; Sobel & Sommerville, 2010). This innate understanding empowers us to recognize that altering specific factors can lead to corresponding outcomes, enabling us to actively manipulate our surroundings to accomplish desired objectives and acquire fresh insights. A deep understanding of cause and effect enables us to explain behaviors (Schult & Wellman, 1997), predict future outcomes (Shultz, 1982), and use counterfactual reasoning to dissect past events (Harris et al., 1996). These cognitive abilities inherently shape human thought and reasoning (Sloman, 2005; Sloman & Lagnado, 2015;

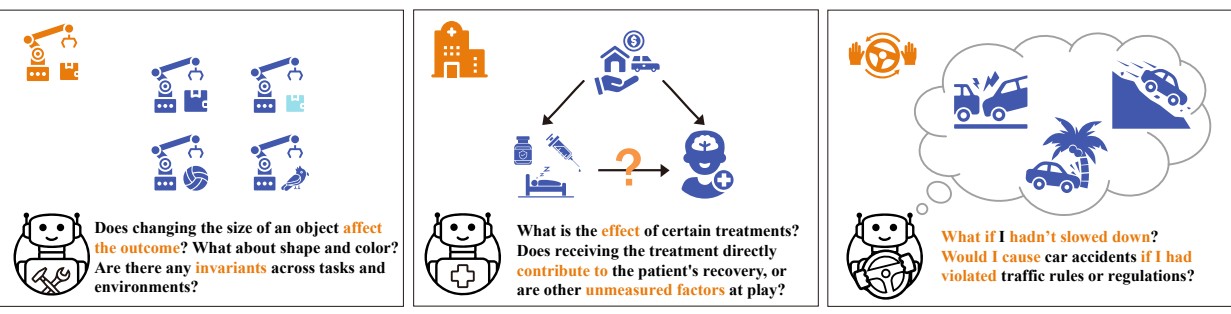

Figure 1: Illustrative examples of causality in reinforcement learning and its impact on decision making.

Pearl, 2009b; Pearl & Mackenzie, 2018), forming the basis for modern society and civilization, as well as propelling advancements in science and technology (Glymour, 1998).

In the pursuit of developing agents with these crucial abilities, reinforcement learning (RL) (Sutton & Barto, 2018) emerges as a promising path. It involves learning optimal decision-making policies by interacting with the environment to learn from the outcome of certain behaviors. This property makes RL naturally connected to causality, as agents can sometimes directly estimate the causal effect for policy-specified interventions. In fact, many important concerns related to causality become pronounced when we examine real-world applications of RL.

Consider the scenarios and the questions presented in Figure 1 where the phrases related to causality are highlighted in orange. In robotic manipulation, RL agents need to understand the effect of altering different factors on the outcomes to avoid learning decisions sensitive to irrelevant factors. Moreover, for effective knowledge transfer, agents are expected to leverage invariance across different tasks and environments. In medical scenarios, which often involve observational studies, agents should be aware of unobserved confounders that may remain hidden in the data generation process, such as socioeconomic status. These confounders can simultaneously affect treatments and outcomes, introducing bias into decision-making by erroneously attributing unobserved factors to treatment, resulting in either overestimation or underestimation of the true treatment effect on outcomes. Additionally, in scenarios like autonomous driving, safety concerns often lead to questions like whether the vehicle would have collided with the leading vehicle if it hadn't slowed down. Answering such queries necessitates comparing the factual world and a hypothetical one in which the vehicle had not slowed down. To address this challenge, agents are expected to utilize the ability of counterfactual reasoning, which empowers them to envision and gain insights from scenarios absent from the collected data, thus improving the efficiency of learning safe driving policies.

These examples underscore the potential of incorporating causality into RL, yet the journey is fraught with challenges. RL agents, lacking a fundamental understanding of the world, typically rely on extensive trial and error to make rational decisions and understand causal relationships among different factors. They may also face difficulties in identifying and learning invariant mechanisms from a limited set of environments and tasks. Moreover, since traditional RL lacks a built-in capacity for modeling causality, agents may be susceptible to spurious correlations, especially in scenarios involving offline learning and partial observability. Addressing these problems necessitates a careful investigation of several critical questions: how to formalize these issues, what assumptions or prior knowledge are required, how to introduce them in a principled manner, and what are the consequences of making incorrect assumptions. Consequently, in recent years, researchers have been actively exploring systematic approaches to integrate causality into the realm of reinforcement learning, giving rise to an emerging field known as causal RL.

Causal RL harnesses the power of causal inference (Pearl, 2009b; Peters et al., 2017), which offers a mathematical framework for formalizing the data generation process and reasoning about causality (Schölkopf et al., 2021; Kaddour et al., 2022). It is an umbrella term for RL approaches that incorporate additional assumptions or knowledge about the underlying causal model to inform decision-making. Modeling causality explicitly not only holds the promise of a principled approach to solving complex decision-making problems

but also enhances the transparency and interpretability of the decision-making process. To further elaborate on the distinctive capabilities of causal RL in addressing the challenges faced by traditional RL, we will delve into a more detailed discussion spanning sections 3 to 6. Note that causal inference, while powerful, is not a panacea; its effectiveness depends on the quality of assumptions and domain expertise applied. We will explore these nuances further in section 7.1 to provide a balanced understanding of the role and limitations of causal RL in addressing the challenges of traditional RL.

Given the successful application of causal inference in various domains such as computer vision (Lopez-Paz et al., 2017; Shen et al., 2018; Tang et al., 2020; Wang et al., 2020b), natural language processing (Wu et al., 2021; Jin et al., 2021; Feder et al., 2022), and recommender systems (Zheng et al., 2021; Zhang et al., 2021b; Gao et al., 2022), it is reasonable to expect that causal RL would help resolve core challenges faced by traditional RL methods and tackle new challenges in increasingly complex application scenarios. Nevertheless, a significant obstacle lies in the lack of a clear and consistent conceptualization of causal assumptions and knowledge, as they have been encoded in diverse forms across prior research, tailored to specific problems and objectives. The use of disparate terminologies and techniques makes it challenging to understand the essence, implications, limitations, and opportunities of causal RL, particularly for those new to the realms of causal inference and RL. In light of this, this paper aims to provide a comprehensive survey of causal RL, consolidating the diverse advancements and contributions within this field, thereby establishing meaningful connections and fostering a cohesive understanding.

Our main contributions to the field are as follows.

- We present a comprehensive survey of causal RL, exploring fundamental questions such as its definition, motivations, and its improvements over traditional RL approaches. Additionally, we provide a clear and concise overview of the foundational concepts in both causality research and RL. To the best of our knowledge, this is the first comprehensive survey of causal RL in the existing RL literature [1].

- We identify the key challenges in RL that can be effectively addressed or improved by explicitly considering causality. To facilitate a deeper understanding of the benefits of incorporating causality-aware techniques, we propose a problem-oriented taxonomy. Furthermore, we conduct a comparative analysis of existing causal reinforcement learning approaches, examining their methodologies and limitations.

- We shed light on promising research directions in causal RL. These include advancing theoretical analyses, establishing benchmarks, and tackling specific learning problems. As these research topics gain momentum, they will propel the application of causal RL in real-world scenarios. Hence, establishing a common ground for discussing these valuable ideas in this burgeoning field is crucial and will foster its continuous development and success.

## 2 Background

To better understand causal RL, an emerging field that combines the strengths of causality research and RL, we start by introducing the fundamentals of and some common concepts relevant to the two research areas.

### 2.1 A Brief Introduction to Causality

We first discuss how to use mathematical language to describe and study causality. In general, there are two primary frameworks that researchers use to formalize causality: SCMs (structural causal models) Pearl (2009a); Glymour et al. (2016) and PO (potential outcome) (Rubin, 1974; Imbens & Rubin, 2015). We focus on the former in this paper because it provides a graphical methodology that can help researchers abstract

---

[1]We note that Schölkopf et al. (2021) and Kaddour et al. (2022) discussed causal RL alongside many other research subjects in their papers. The former mainly studied the causal representation learning problem, and the latter comprehensively investigated the field of causal machine learning. The present study, however, focuses on examining the literature on causal RL and provides a systematic review of the field.

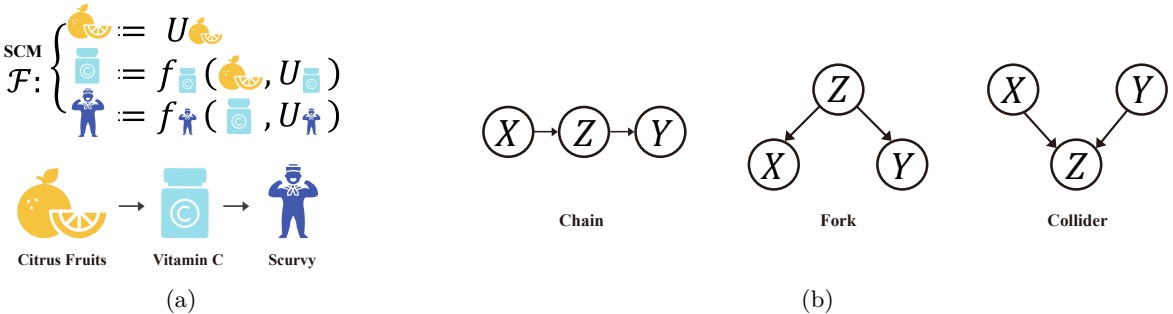

(a)                                                        (b)

Figure 2: (a) The SCM and causal graph for the scurvy example, in which consuming fruits helps to prevent scurvy by influencing the intake of vitamin C. (b) The three basic building blocks of causal graphs.

and better understand the data generation process. It is noteworthy that these two frameworks are logically equivalent, and most assumptions are interchangeable.

**Definition 2.1** (Structural Causal Model). An SCM $\mathcal{M}$ is represented by a tuple $(\mathcal{V}, \mathcal{U}, \mathcal{F}, P(\mathbf{U}))$, where

- $\mathcal{V} = \{V_1, V_2, \cdots, V_m\}$ is a set of endogenous variables that are of interest in a research problem,

- $\mathcal{U} = \{U_1, U_2, \cdots, U_n\}$ is a set of exogenous variables that represent the source of stochasticity in the model and are determined by external factors that are generally unobservable,

- $\mathcal{F} = \{f_1, f_2, \cdots, f_m\}$ is a set of structural equations that assign values to each of the variables in $\mathcal{V}$ such that $f_i$ maps $\mathbf{PA}(V_i) \cup U_i$ to $V_i$, where $\mathbf{PA}(V_i) \subseteq \mathcal{V} \backslash V_i$ and $U_i \subseteq \mathcal{U}$,

- $P(\mathbf{U})$ is the joint probability distribution of the exogenous variables in $\mathcal{U}$.

**Structural causal model** provides a rigorous framework for examining how relevant features of the world interact. Each structural equation $f_i \in \mathcal{F}$ specifies the value of an endogenous variable $V_i$ based on an exogenous variable $U_i$ and $\mathbf{PA}(V_i)$. These equations establish the causal links between variables and mathematically characterize the underlying mechanisms of the data generation process. If all $U_i \in \mathcal{U}$ are independent of each other, it follows that each endogenous variable $V_i \in \mathcal{V}$ is independent of all its nondescendants, given its parents $\mathbf{PA}(V_i)$. In this case, we refer to the model as *Markovian* and $\mathbf{PA}(V_i)$ is considered complete, as it includes all the relevant immediate causes of $V_i$. This assumption is prevalent in human discourse, probably because it allows us to omit some causes from $\mathbf{PA}(V_i)$ (which are aggregated as exogenous variables and summarized by probabilities), forming a useful abstraction of the underlying physical processes that might be overly detailed for practical use (Pearl, 2009b, Chapter 2).

**Causal graph**. Each SCM $\mathcal{M}$ is associated with a causal graph $\mathcal{G} = \{\mathcal{V}, \mathcal{E}\}$, where nodes $\mathcal{V}$ represent endogenous variables and edges $\mathcal{E}$ represent causal relationships determined by the structural equations. Specifically, an edge $e_{ij} \in \mathcal{E}$ from node $V_j$ to node $V_i$ exists if the random variable $V_j \in \mathbf{PA}(V_i)$. In some cases, there may be causes that are omitted from $\mathbf{PA}(V_i)$ but affect more than one variable in $\mathcal{V}$. These omitted variables are referred to as *unobserved confounders*. In such cases, exogenous variables are not independent of each other, which leads to the loss of Markov property. If we explicitly treat such variables as latent variables and represent them with nodes in the graph, the Markov property is restored (Pearl, 2009b, Chapter 2). Figure 2a illustrates an SCM and its corresponding causal graph. This example includes three binary endogenous variables - consumption of fruits, intake of vitamin C, and occurrence of scurvy - along with the relevant exogenous variables. In this example, the consumption of fruits does not directly protect the health of sailors from scurvy; instead, it produces an indirect effect by influencing the intake of vitamin C. Therefore it is not part of the structural equation that determines sailors' health. Figure 2b introduces the three fundamental building blocks of the causal graph: chain, fork, and collider. These simple structures can be combined together to create more complex data generation processes.

**Product decomposition** involves representing a complex joint probability distribution $P(\mathbf{V})$ as a product of conditional probability distributions that are easier to model and analyze. By applying the chain rule, we

can always decompose $P(\mathbf{V})$ of $n$ variables $V_1, \cdots, V_n \in \mathcal{V}$ as a product of $n$ conditional distributions:

$$P(\mathbf{V}) = \prod_{i=1}^{n} P(V_i|V_1, \cdots, V_{i-1}).$$

While this decomposition method is general, it does not consider the causal relationships within the data. Therefore, predecessor variables $V_1, \cdots, V_{i-1}$ are not necessarily the causes of $V_i$ and the conditional probability of $V_i$ is not necessarily sensitive to all predecessor variables. Assuming that the data is generated by a Markovian causal model, the causal Markov condition (Pearl, 2009b, Chapter 1) helps us establish a connection between causation and probabilities, thereby allowing for a more parsimonious decomposition. This condition states that for every Markovian causal model with a causal graph $\mathcal{G}$, the induced joint distribution $P(\mathbf{V})$ is Markov relative to $\mathcal{G}$. Specifically, a variable $V_i \in \mathcal{V}$ is independent of any non-descendants given its parents $\mathrm{PA}(V_i)$ in $\mathcal{G}$. This property enables a structured decomposition along causal directions, which is referred to as the causal factorization (or the disentangled factorization) (Schölkopf et al., 2021):

$$P(\mathbf{V}) = \prod_{i=1}^{n} P(V_i|\mathrm{PA}(V_i)), \tag{1}$$

where the (conditional) probability distributions of the form $P(V_i|\mathrm{PA}(V_i))$ are commonly referred to as *causal Markov kernels* (Peters et al., 2017) or *causal mechanisms* (Schölkopf et al., 2021). While equation 1 is not the only method for product decomposition, it is the only one that decomposes $P(\mathbf{V})$ as the product of causal mechanisms. For further information regarding the product decomposition of semi-Markovian models, in which the causal graph is acyclic but the exogenous variables are not jointly independent, please refer to Bareinboim et al. (2022).

**Intervention**. SCMs not only offer a rigorous mathematical framework for studying causal relationships but also facilitate the modeling of external interventions. Specifically, an intervention can be encoded as an alteration of some of the structural equations in $\mathcal{M}$. For example, forcing the variable $X = x$ forms a submodel $\mathcal{M}_x$, in which the set of structural equations $\mathcal{F}_x = \{f_i : V_i \notin X\} \cup \{X = x\}$. To predict the outcome of such an intervention, we simply employ the submodel $\mathcal{M}_x$ to compute the new probability distribution. In addition to directly setting the variables to constant values (known as hard intervention), we can also manipulate the probability distributions that govern the variables, preserving some of the original dependencies (known as soft intervention). Many research questions involve estimating the effects of interventions. For example, preventing scurvy requires identifying effective interventions (e.g., through dietary changes) that reduce the probability of getting scurvy. To distinguish from conditional probability, researchers introduced the do-operator, using $P(Y|\mathrm{do}(X = x))$ to denote the intervention probability, meaning the probability distribution of the outcome variable $Y$ when $X$ is fixed to $x$. Figure 3a illustrates the difference between conditional and intervention probabilities. It is noteworthy that a critical distinction between statistical models and causal models lies in the fact that the former specifies a single probability distribution, while the latter represents a collection of distributions, one for each possible intervention (including the null intervention). Consequently, causal models can provide a more comprehensive understanding of the world, potentially enhancing an agent's robustness against certain distribution shifts (Schölkopf et al., 2021; Thams, 2022).

**Counterfactual**. Counterfactual thinking involves posing hypothetical questions, such as, "Would the scurvy patient have stayed healthy if they had eaten enough fruit". This cognitive process allows us to retrospectively analyze past events and reason about the potential outcomes of altering certain factors in the past. This type of thinking helps us gain insights from experiences and identify opportunities for further improvements. Since counterfactuals involve hypothetical scenarios, collecting counterfactual data is generally impossible in reality, which is the core difference between it and interventions.

In the context of SCMs, counterfactual variables are often denoted with a subscript, such as $Y_{X=x}$ (or $Y_x$ when there is no ambiguity) where $X$ and $Y$ are two sets of variables in $\mathcal{V}$. This notation helps researchers differentiate counterfactuals from the original variable $Y$. The key difference between $Y$ and $Y_x$ is that the latter is generated by a submodel $\mathcal{M}_x$ in which $X$ is set to $x$. Figure 3b provides an illustration for evaluating counterfactuals with the twin network method (Pearl, 2009b, Chapter 7). The two networks represent the

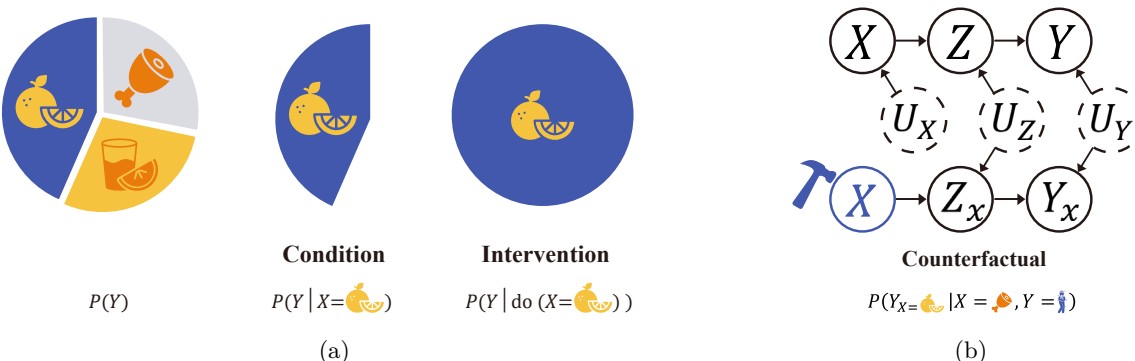

Figure 3: (a) An illustration of the difference between condition and intervention, with $Y$ representing the health of sailors and $X$ being a discrete variable with three possible values: fruit, juice, or meat. The marginal probability distribution $P(Y)$ studies all subgroups within the population, while the conditional probability distribution $P(Y|X = \text{fruit})$ focuses on the subgroup of sailors who have consumed fruits. In contrast, the interventional probability distribution $P(Y|\text{do}(X = \text{fruit}))$ examines the population in which all sailors are required to consume fruits. (b) An illustration of counterfactual probabilities, which study events occurring in an imaginary world (the bottom network), e.g., considering the sick sailors who ate meat in the factual world (the upper network), would they have stayed healthy if they had consumed fruits?

factual and counterfactual (imaginary) world respectively. They are connected by the exogenous variables, sharing the same structure and variables of interest, except the counterfactual one removes arrows pointing to the variables corresponding to the hypothetical interventions. Remark that there are many different types of counterfactual quantities implied by a causal model other than the one shown in this example. In some scenarios, such as fairness analysis (Carey & Wu, 2022; Plecko & Bareinboim, 2022), we may need to study nested counterfactuals, e.g., the direct and indirect effects.

Together, correlation, intervention, and counterfactuals form the three rungs of the ladder of causation, also known as the "Pearl Causal Hierarchy" (Bareinboim et al., 2022), which naturally emerges from an SCM. This hierarchy involves increasingly refined reasoning tasks, as does the knowledge required to complete these tasks. In summary, we have provided a concise overview of several fundamental concepts in this section. For readers interested in delving deeper into the realm of causality, please refer to Appendix A and the cited references. Subsequent sections will discuss related concepts and techniques within the context of reinforcement learning.

## 2.2 A Brief Introduction to Reinforcement Learning

Reinforcement learning studies sequential decision problems. Mathematically, we can formalize these problems as Markov decision processes.

**Definition 2.2** (Markov decision process). An MDP $\mathcal{M}$ is specified by a tuple $\{\mathcal{S}, \mathcal{A}, P, R, \mu_0, \gamma\}$, where

- $\mathcal{S}$ denotes the state space and $\mathcal{A}$ denotes the action space,

- $P : \mathcal{S} \times \mathcal{A} \times \mathcal{S} \to [0, 1]$ is the transition probability function that yields the probability of transitioning into the next states $s_{t+1}$ after taking an action $a_t$ at the current state $s_t$,

- $R : \mathcal{S} \times \mathcal{A} \to \mathbb{R}$ is the reward function that assigns the immediate reward for taking an action $a_t$ at state $s_t$,

- $\mu_0 : \mathcal{S} \to [0, 1]$ is the probability distribution that specifies the generation of the initial state, and

- $\gamma \in [0, 1]$ denotes the discount factor that accounts for how much future events lose their value as time passes.

**Markov decision processes**. In definition 2.2, the decision process starts by sampling an initial state $s_0$ with $\mu_0$. An agent takes responsive action using its policy $\pi$ (a function that maps a state to an action) and receives a reward from the environment assigned by $R$. The environment evolves to a new state following $P$; then, the agent senses the new state and repeats interacting with the environment. The goal of an RL agent is to search for the optimal policy $\pi^*$ that maximizes the return (cumulative reward) $G_0$. In particular, at any timestep $t$, the return $G_t$ is defined as the sum of discounted future rewards, i.e., $G_t = \sum_{i=0}^{\infty} \gamma^i R_{t+i}$. A multi-armed bandit (MAB) is a special type of MDP that focuses on single-step decision-making problems. On the other hand, a partially observable Markov decision process (POMDP), generalizes the scope of MDPs by considering partial observability. In a POMDP, the system still operates based on an MDP, but the agent can only access a partial or incomplete description of the system state, often referred to as the observation $O_t$, when making decisions. For example, in a video game, a player may need to deduce the motion of a dynamic object based on the visual cues displayed on the current screen.

**Value functions**. The return $G_t$ evaluates how good an action sequence is. However, in stochastic environments, the same action sequence can lead to diverse trajectories and consequently, different returns. Moreover, a stochastic policy $\pi$ outputs a probability distribution over the action space. Considering these stochastic factors, the return $G_t$ associated with a policy $\pi$ becomes a random variable. In order to evaluate policies under uncertainty, RL introduces the concept of value functions. There are two types of value functions: $V^\pi(s)$ denotes the expected return obtained by following the policy $\pi$ from state $s$; $Q^\pi(s,a)$ denotes the expected return obtained by performing action $a$ at state $s$ and following the policy $\pi$ thereafter. The optimal value functions correspond to the optimal policy $\pi^*$ are denoted by $V^*(s)$ and $Q^*(s,a)$.

**Bellman equations**. By definition, $V^\pi(s) = \mathbb{E}_\pi[G_t|S_t = s]$ and $Q^\pi(s,a) = \mathbb{E}_\pi[G_t|S_t = s, A_t = a]$. These two types of value functions can be expressed in terms of one another. By expanding the return $G_t$, we can rewrite value functions in a recursive manner:

$$
\begin{aligned}
V^\pi(s) &= \sum_{a \in \mathcal{A}} \pi(a|s) \left( R(s,a) + \gamma \sum_{s' \in \mathcal{S}} P(s'|s,a) V^\pi(s') \right) \\
Q^\pi(s,a) &= R(s,a) + \gamma \sum_{s' \in \mathcal{S}} \sum_{a' \in \mathcal{A}} \pi(a'|s') Q^\pi(s',a').
\end{aligned}
\tag{2}
$$

When the timestep $t$ is not specified, $s$ and $s'$ are often used to refer to the states of two adjacent steps. The above equations are known as the Bellman expectation equations, which establish the connection between two adjacent steps. Similarly, the Bellman optimality equations relate the optimal value functions:

$$
\begin{aligned}
V^*(s) &= \max_{a \in \mathcal{A}} \left( R(s,a) + \gamma \sum_{s' \in \mathcal{S}} P(s'|s,a) V^*(s') \right) \\
Q^*(s,a) &= R(s,a) + \gamma \sum_{s' \in \mathcal{S}} P(s'|s,a) \max_{a' \in \mathcal{A}} Q^*(s',a').
\end{aligned}
\tag{3}
$$

When the environment (also referred to as the dynamic model or, simply, the model) is known, the learning problem simplifies into a planning problem that can be solved using dynamic programming techniques based on the Bellman equations. However, in the realm of RL, the main focus is on unknown environments. In other words, agents do not possess complete knowledge of the transition function $P(s'|s,a)$ and the reward function $R(s,a)$. This characteristic brings RL closer to decision-making problems in real-world scenarios.

**Categorizing reinforcement learning methods**. There are several ways to categorize RL methods. One approach is based on the agent's components. Policy-based methods generally focus on optimizing an explicitly parameterized policy to maximize the return, while value-based methods use collected data to fit a value function and derive the policy implicitly from it. Actor-critic methods combine both of them, equipping an agent with both a value function and a policy. Another classification criterion is whether RL methods use an environmental model. Model-based reinforcement learning (MBRL) methods typically employ a well-defined environmental model (such as AlphaGo (Silver et al., 2017)) or construct one using the collected data. The model assists the agent in planning or generating additional training data, thereby enhancing the learning process. Furthermore, RL can also be divided into on-policy, off-policy, and offline

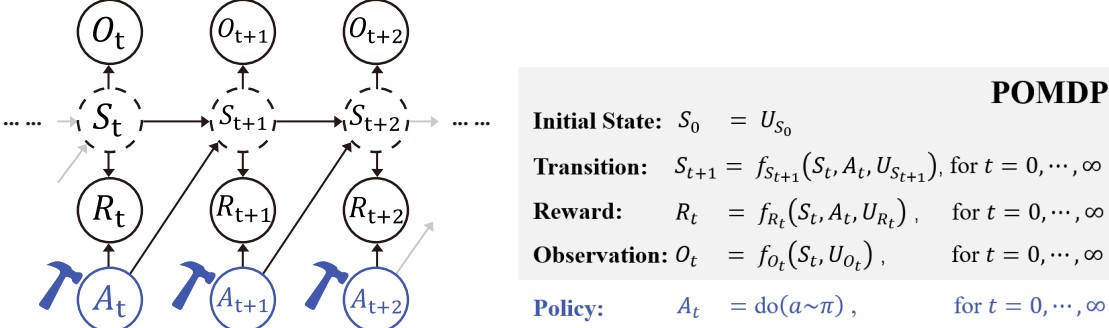

Figure 4: An illustrative example of casting a POMDP problem into an SCM. States are marked with dashed circles to emphasize that they are latent variables in the POMDP problem, while actions are marked with hammers as they represent intervention variables controlled by policies.

approaches based on data collection. On-policy RL only utilizes data from the current policy, while off-policy RL involves data collected by other policies. Offline RL disallows data collection, restricting the agent to learn from a fixed dataset.

## 2.3 Causal Reinforcement Learning

Before formally defining causal RL, let us cast a POMDP problem into an SCM $\mathcal{M}$. To do this, we consider the observation, state, action, and reward at each step to be endogenous variables. The observation, state transition, and reward functions are then described as deterministic functions with independent exogenous variables, represented by a set of structural equations $\mathcal{F}$ in $\mathcal{M}$. This transformation is always possible using autoregressive uniformization (Buesing et al., 2019), without imposing any extra constraints. It allows us to formally discuss causality in RL, including addressing counterfactual queries that cannot be explained by non-causal methods. Figure 4 presents an illustrative example of this transformation. From this formalization, it is straightforward to derive the MDP regime by introducing an additional constraint: $O_t = S_t$. In practice, states and actions may have high dimensionality, and the granularity of the causal model can be adjusted based on our prior knowledge. While the SCM representation allows us to reason about causality in decision-making problems and organize causal knowledge in a clear and reusable way, it does not constitute causal RL on its own. In this paper, we define causal RL as follows.

**Definition 2.3** (Causal reinforcement learning)**.** Causal RL is an umbrella term for RL approaches that incorporate additional assumptions or prior knowledge to analyze and understand the causal mechanisms underlying actions and their consequences, enabling agents to make more informed and effective decisions.

This definition emphasizes two fundamental aspects that distinguish causal RL from non-causal RL. 1) It emphasizes a focus on causality, seeking to advance beyond superficial associations or data patterns. To meet this goal, 2) it necessitates the incorporation of additional assumptions or knowledge that accounts for the causal relationships inherent in decision-making problems.

The primary objective of RL is to determine the policy $\pi$ that yields the highest expected return, rather than inferring the causal effect of a specific intervention. The policy $\pi$ can be seen as a soft intervention that preserves the dependence of the action on the state, i.e., $\text{do}(a \sim \pi(\cdot|s))$. Different policies result in varying trajectory distributions. As mentioned earlier, RL is close to causality because on-policy RL can directly learn the total effects of actions on the outcome from interventional data. However, when we reuse observational data, as in off-policy/offline RL, the learning problem becomes more intricate, as agents may suffer from spurious correlations (Zhang et al., 2020b; Deng et al., 2021). Additionally, we may be interested in certain types of counterfactual quantities other than total effects in causal RL, as they hold the promises of improving data efficiency and performance (Bareinboim et al., 2015; Buesing et al., 2019; Lu et al., 2020).

We note that there is a lack of clarity and coherence in the existing literature on causal RL, primarily because causal modeling is more of a mindset than a specific problem setting or solution. Previous work has explored diverse forms of causal modeling, driven by different prior knowledge and research purposes. Ideally, a perfect understanding of the data generation process would grant access to the true causal model, enabling us to answer any correlation, intervention, and even counterfactual inquiries. However, given the inherent complexity of the world, it is often impractical to access a fully specified SCM. Most of the time, we can only access data generated by this model. Unfortunately, data alone is generally insufficient to overcome the knowledge gap. The Causal Hierarchy Theorem (CHT) (Bareinboim et al., 2022) demonstrates that the ability to address questions at one layer almost never guarantees the ability to address the questions at higher layers. Fortunately, certain forms of knowledge, such as causal graphs, may provide sufficient but attainable insights into the underlying model. They can serve as valuable surrogates, enabling us to identify the quantities of interest as if the SCM were accessible. Note that, inferring an interventional distribution from the observational distribution and the causal graph may not always be feasible due to the presence of unobserved confounders. For more information about the identifiability issue, we refer to Bareinboim et al. (2022). Additionally, in scenarios involving multiple domains, it is often beneficial to examine invariant factors across these domains, including causal mechanisms, causal structure (causal graph), and causal representation (high-level variables that capture the causal mechanisms underlying the data). In cases where prior knowledge about these factors is lacking, we can introduce certain inductive biases, such as sparsity, independent causal mechanisms, and sparse mechanism shifts, to obtain reasonable estimates Schölkopf et al. (2021); Sontakke et al. (2021); Huang et al. (2022a;b).

With a solid understanding of the foundational concepts and definitions, we are now well-equipped to explore the realm of causal reinforcement learning. The upcoming sections delve into four crucial challenges where causal RL demonstrates its potential: generalizability and knowledge transfer, spurious correlations, sample efficiency, and considerations beyond return, e.g., explainability, fairness, and safety.

## 3 Advancing Generalizability and Knowledge Transfer through Causal Reinforcement Learning

### 3.1 The Issue of Generalizability in Reinforcement Learning

Generalizability poses a major challenge in the deployment of RL algorithms in real-world applications. It refers to the ability of a trained policy to perform effectively in new and unseen situations (Kirk et al., 2022). The issue of training and testing in the same environment has long been a critique faced by the RL community (Irpan, 2018). While people often expect RL to work reliably in different (but similar) environments or tasks, traditional RL algorithms are typically designed for solving a single MDP. They can easily overfit the environment, failing to adapt to minor changes. Even in the same environment, RL algorithms may produce widely varying results with different random seeds (Zhang et al., 2018a;b), indicating instability and overfitting. Lanctot et al. (2017) presented an example of overfitting in multi-agent scenarios in which a well-trained RL agent struggles to adapt when the adversary slightly changes its strategy. A similar phenomenon was observed by Raghu et al. (2018). Furthermore, considering the non-stationarity and constant evolution of the real world (Hamadanian et al., 2022), there is a pressing need for robust RL algorithms that can effectively handle changes. Agents should possess the capability to transfer their acquired skills across varying situations rather than relearning from scratch.

### 3.2 How can Causality Help to Improve Generalization and Facilitate Knowledge Transfer?

Some previous studies have shown that data augmentation improves generalization (Lee et al., 2020a; Wang et al., 2020a; Yarats et al., 2021), particularly for vision-based control. This process involves generating new data by randomly shifting, mixing, or perturbing observations, which makes the learned policy more resistant to irrelevant changes. Another common practice is domain randomization. In sim-to-real reinforcement learning, researchers randomized the parameters of simulators to facilitate adaptation to reality (Tobin et al., 2017; Peng et al., 2018). Additionally, some approaches have attempted to incorporate inductive bias

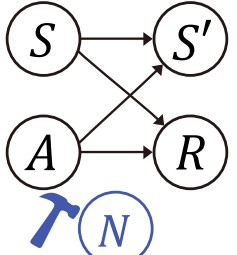 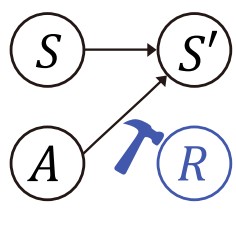 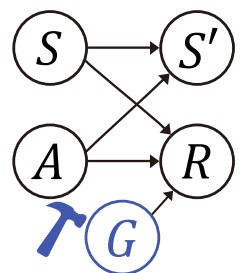 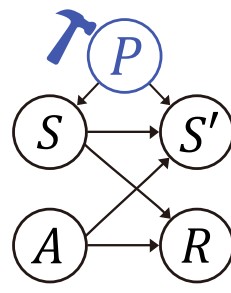

(a) Generalize to noise or ir-  (b) Generalize to different  (c) Generalize to different  (d) Generalize to different
relevant variables.          reward assignments.          goals.                      physical properties.

Figure 5: Different types of generalization problems in reinforcement learning represented by causal graphs. In these graphs, $S$ and $S'$ represent states in adjacent time steps, $A$ represents actions, $R$ represents rewards, $N$ represents irrelevant variables (e.g., background color), $G$ represents goals (e.g., target position), and $P$ represents physical properties (e.g., mass).

by designing special network structures to improve generalization performance (Kansky et al., 2017; Higgins et al., 2017; Zambaldi et al., 2019; Raileanu & Fergus, 2021).

While these works have demonstrated empirical success, explaining why certain techniques outperform others remains challenging. This knowledge gap hinders our understanding of the underlying factors that drive successful generalization and the design of algorithms that reliably generalize in real-world scenarios. To tackle this challenge, it is essential to identify the factors that drive changes. Kirk et al. (2022) proposed using contextual MDP (CMDP) (Hallak et al., 2015) to formalize generalization problems in RL. A CMDP resembles a standard MDP but explicitly captures the variability across a set of environments or tasks, which is determined by contextual variables such as goals, colors, shapes, mass, or the difficulty of game levels. From a causal perspective, these variabilities can be interpreted as different types of external interventions in the data generation process (Schölkopf et al., 2021; Thams, 2022). Figure 5 illustrates some examples of the generalization problems corresponding to different interventions. Previous methods simulate these interventions during the training phase by augmenting original data or randomizing certain attributes, allowing the model to learn from various domains. By carefully scrutinizing the causal relationships behind the data, we can gain a better understanding of the sources of generalization ability and provide a more logical explanation.

More importantly, by making explicit assumptions on what changes and what remains invariant, we can derive principled methods for effective knowledge transfer and adaptation (Zhang et al., 2015; Gong et al., 2016). To illustrate this point, let us recall the example discussed in Figure 2a, where we can use $X$, $Y$, and $Z$ to represent fruit consumption, vitamin C intake, and the occurrence of scurvy, respectively. Consider an intervention on fruit consumption (the variable $X$), one would have to retrain all modules in a non-causal factorization such as $P(X, Y, Z) = P(Y)P(Z|Y)P(X|Y, Z)$ due to the change in $P(X)$. In contrast, with the causal factorization $P(X, Y, Z) = P(X)P(Z|X)P(Y|Z)$, only $P(X)$ needs to be adjusted to fit the new domain. The intuition behind this example is quite straightforward: altering fruit consumption ($P(X)$) does not impact the vitamin C content in specific fruits ($P(Z|X = x)$) or the likelihood of developing scurvy conditioned on the amount of vitamin C intake ($P(Y|Z = z)$). This property is referred to as *independent causal mechanisms* (Schölkopf et al., 2021), indicating that the causal generation process comprises stable and autonomous modules (causal mechanisms) (Pearl, 2009b, Chapter 2) such that changing one does not affect the others. Building on this concept, the sparse mechanism shift hypothesis (Schölkopf et al., 2021; Perry et al., 2022) suggests that small changes in the data distribution generally correspond to changes in only a subset of causal mechanisms. These assumptions provide a basis for designing efficient algorithms and models for knowledge transfer.

Table 1: Selected methods utilizing causality to improve generalizability.

| Category | Paper | Techniques | Settings | Environments or Tasks |
|---|---|---|---|---|
| Irrelevant variables | Zhang et al. (2020a) | Causal representation learning | online | Toy [2] |
| | | | | Cart-pole (dm_control) |
| | | | | Cheetah (dm_control) |
| | Bica et al. (2021b) | Causal representation learning | imitation | OpenAI Gym |
| | | | | MIMIC III |
| | Wang et al. (2022) | Causal dynamics learning | online* | Chemical |
| | | | | Manipulation (robosuite) |
| | Saengkyongam et al. (2022) | Causal representation learning | offline | Toy |
| | Ding et al. (2022) | Causal discovery | online | Manipulation (Not accessible) |
| | | | | Unlock (Minigrid) |
| | | | | Crash (highway-env) |
| Dynamics | Sontakke et al. (2021) | Causal representation learning | offline-to-online | Manipulation (CausalWolrd) |
| | Lee et al. (2021) | Intervention | online | Manipulation (Isaac Gym) |
| | | Domain randomization | | |
| | Zhu et al. (2021) | Counterfactual reasoning | online | Manipulation (CausalWolrd) |
| | Guo et al. (2022b) | Mediation analysis | online | Pendulum (OpenAI Gym) |
| | | | | Locomotion (OpenAI Gym) |
| Tasks | Eghbal-zadeh et al. (2021) | Causal representation learning | online | Contextual-Gridworld |
| | Pitis et al. (2022) | Counterfactual reasoning | offline | Spriteworld |
| | | | | Pong (Roboschool) |
| | | | | Manipulation (OpenAI) |
| Dynamics and Tasks | Zhang & Bareinboim (2017) | Causal bound [3] | online* | Toy |
| | Dasgupta et al. (2018) | Meta learning | online | Toy |
| | Nair et al. (2019) | Causal induction [4] | online | Light |
| | Huang et al. (2022a) | Causal dynamics learning | online | Cart Pole (OpenAI Gym) |
| | | | | Pong (OpenAI Gym) |
| Others | Zhu et al. (2022b) | Causal discovery | offline | Toy |
| | | Causal dynamics learning | | Inverted Pendulum (OpenAI Gym) |

## 3.3 Causal Reinforcement Learning for Improving Generalizability

Generalization involves various settings. Zero-shot generalization entails the agent solely acquiring knowledge in training environments and then being evaluated in unseen scenarios. While this setting is appealing, it is often impractical in real-world scenarios. Alternatively, we may allow agents to receive additional training in target domains, categorized as transfer RL (Zhu et al., 2020), multitask RL (Vithayathil Varghese & Mahmoud, 2020), lifelong RL (Khetarpal et al., 2022), among others. omprehend the essential capabilities required for generalization and the expected outcomes of learning algorithms, causal RL explicitly considers the factors that govern changes in distribution. This section, therefore, categorizes existing causal RL approaches for generalization based on specific factors of change. The representative works are shown in Table 1. Furthermore, we also label the problem settings for these approaches, including offline, online, offline-to-online, and imitation learning. In cases where an approach employs online training augmented by offline datasets, it is labeled as online*. To be self-contained, we offer a concise overview of the environments and tasks in Appendix B.

### 3.3.1 Generalize to Different Environments

First, we consider how to generalize to different environments. From a causal perspective, different environments share most of the causal mechanisms but differ in certain modules, resulting from different interventions in the state or observation variables. Building on the causal relationships within these variables, we can categorize existing approaches into two main groups: generalization to irrelevant variables and generalization to varying dynamics.

---

[2] The term "toy" refers to simple, synthetically constructed datasets or simulation environments that are used to experimentally verify findings. It is not a concrete environment or task. We use this term consistently throughout the paper.

[3] When the causal effect is unidentifiable, we can resort to set identification (partial identification), deriving its upper and lower bounds, which allows us to assess the robustness of our estimates against unobserved confounding.

[4] Causal induction involves the process of extracting abstract causal variables from high-dimensional, low-level pixel representations, followed by recovering the underlying causal graph.

To enhance the ability to generalize to irrelevant factors, RL agents must examine the causality to identify the invariance in the data generation process. Zhang et al. (2020a) investigated the problem of generalizing to diverse observation spaces within the block MDP framework, such as robots equipped with different types of cameras and sensors, which is a common scenario in reality. In the block MDP framework, the observation space may be infinite, but we can uniquely determine the state (finite but unobservable) given the observation. The authors proposed using invariant prediction to learn the causal representation that generalizes to unseen observations. Similarly, Bica et al. (2021b) introduced invariant causal imitation learning, which learns the imitation policy based on invariant causal representation across multiple environments. Wang et al. (2022) studied the causal dynamics learning problem, which attempts to eliminate irrelevant variables and unnecessary dependencies so that policy learning will not be affected by these nuisance factors. Saengkyongam et al. (2022) focused on the offline contextual bandit problems. Their proposed approach involves iteratively assessing the invariance condition for various subsets of variables to learn an optimal invariant policy. The algorithm begins by generating a sample dataset using an initial policy and then tests the invariance of each subset across different environments. If a subset is found to be invariant, an optimal policy within that subset is learned through off-policy optimization. The experimental results suggest that invariance is crucial for obtaining distributionally robust policies, particularly in the presence of unobserved confounders. Ding et al. (2022) proposed an approach to address the generalization problem in goal-conditioned reinforcement learning (GCRL). Their method involves treating the causal graph as a latent variable and optimizing it using a variational likelihood maximization procedure. This method trains agents to discover causal relationships and learn a causality-aware policy that is robust against changes in irrelevant variables.

Generalizing to new dynamics is a complex issue that involves different types of variations. These variations may include changes in physical properties (e.g., gravitational acceleration), disparities between the simulation environment and reality, and alternations in the range of attribute values, etc. Sontakke et al. (2021) proposed training RL agents to infer and categorize causal factors in the environment with experimental behavior learned in a self-supervised manner. These behaviors can help the agent to extract discrete causal representations from collected trajectories, which can be applicable to unseen environments, empowering the agent to effectively generalize to unseen contexts. Lee et al. (2021) proposed an approach that conducts interventions to identify relevant state variables for successful robotic manipulation, i.e., the features that causally influence the outcome. The robot exhibited excellent sim-to-real generalizability after training with domain randomization on the identified features. Zhu et al. (2021) developed an algorithm to improve the ability of agents to generalize to rarely seen or unseen object properties. This algorithm models the environmental dynamics with SCMs, allowing the agent to generate counterfactual trajectories about objects with different attribute values, which leads to improved generalizability. Guo et al. (2022b) investigated the unsupervised dynamics generalization problem, which allows the learned model to generalize to new environments. The authors approached this challenge by leveraging the intuition that data originating from the same trajectory or similar environments should have similar properties (hidden variables encoded from trajectory segments) that lead to similar causal effects. To measure similarity, they employed conditional direct effects in mediation analysis. The experimental results show that the learned model performs well in new environmental dynamics.

### 3.3.2 Generalize to Different Tasks

Another important topic is how to generalize to different tasks. In the SCM framework, different tasks are created by altering the structural equation of the reward variable or its parent nodes on the causal graph. These tasks have the same underlying environmental dynamics, but the rewards are assigned differently.

Eghbal-zadeh et al. (2021) introduced causal contextual RL, where agents aim to learn adaptive policies that can effectively adapt to new tasks defined by contextual variables. The authors proposed a contextual attention module that enables agents to incorporate disentangled features as contextual factors, leading to improved generalization compared to non-causal agents. In order to make RL more effective in complex, multi-object environments, Pitis et al. (2022) suggested factorizing the state-action space into separate local subsets. This approach allows for learning the causal dynamic model as well as generating counterfactual

transitions in a more efficient manner. By training agents on counterfactual data, the proposed algorithm exhibits improved generalization to out-of-distribution tasks.

Furthermore, in reality, generalization may involve changes in both the environmental dynamics and the task. Several studies have explored this problem from a causal viewpoint. Zhang & Bareinboim (2017) investigated knowledge transfer across bandit agents in scenarios where causal effects are unidentifiable. The proposed strategy combines two steps: deriving the upper and lower bounds of causal effects using structural knowledge and then incorporating these bounds in a dynamic allocation procedure to guide the search toward more promising actions in new bandit problems. The results indicated that this strategy dominates previously known algorithms and achieves faster convergence rates. Dasgupta et al. (2018) explored whether the ability to perform causal reasoning emerges from meta-learning on a simple domain with five variables. The experimental results suggested that the agents demonstrated the ability to conduct interventions and make sophisticated counterfactual predictions. These emergent abilities can effectively generalize to new causal structures. Nair et al. (2019) studied the causal induction problem with visual observation. They incorporated attention mechanisms into the agent to generate a causal graph based on visual observations and use it to make informed decisions. The experiments demonstrated that the agent effectively generalizes to new tasks and environments with unknown causal structures. More recently, Huang et al. (2022c) proposed AdaRL, a novel framework for adaptive RL that learns a latent representation with domain-shared and domain-specific components across multiple source domains. The latent representation is then used in policy optimization. This framework allows for efficient policy adaptation to new environments, tasks, or observations, by estimating new domain-specific parameters using a small number of samples.

### 3.3.3 Other Generalization Problems

In offline RL, the agent can only learn from pre-collected datasets. In this setting, agents may encounter previously unseen state-action pairs during the testing phase, leading to the distributional shift issue (Levine et al., 2020). Most existing approaches mitigate this issue through conservative or pessimistic learning (Fujimoto et al., 2019; Kumar et al., 2020; Yang et al., 2021b), rarely considering generalization to new states. Zhu et al. (2022b) proposed a solution to generalize to unseen states. They recovered the causal structure from offline data by using causal discovery techniques, and then employ an offline MBRL algorithm to learn from the causal model. The experimental results suggested that the causal world model exhibits better generalization performance than a traditional world model, and effectively facilitates offline policy learning.

At the end of this section, it is worth noting that in the field of causal inference, there exists a closely related concept known as transportability. Specifically, transportability focuses on extrapolating experimental findings across domains, i.e., transferring causal effects learned in experimental studies to new domains (populations/environments) in which only observational studies are feasible. Researchers have developed graphical methods and a complete algorithmic program to address this challenge (Pearl & Bareinboim, 2014). Given sufficient structured knowledge (which can qualitatively determine the differences between the two populations), this algorithmic program can help us determine whether the causal effects of the target population can be inferred from the experimental findings of the source population. Furthermore, it can elucidate what experimental and observational findings from the two populations are essential for such an inference. This setting has received limited attention in the causal RL literature, whereas many studies focus on settings that allow experiments (online data collection) to be conducted in multiple domains or use observational data to enhance online learning. Note that this problem is a very prevalent and important aspect of scientific investigations. As researchers in both fields engage in more active knowledge exchange, we believe new and valuable research directions will emerge. For further information on transportability, please refer to Pearl & Bareinboim (2014); Bareinboim & Pearl (2016).

## 4 Addressing Spurious Correlations through Causal Reinforcement Learning

### 4.1 The Issue of Spurious Correlation in Reinforcement Learning

Making reliable decisions based solely on data is inherently challenging, as correlation does not necessarily imply causation. It is important to recognize the presence of spurious correlations, which are deceptive

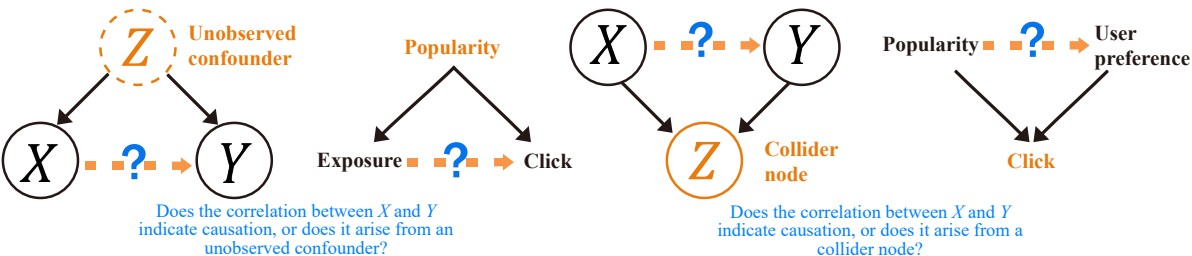

Figure 6: Causal graphs illustrating the two types of spurious correlations, with examples from real-world applications.

associations between two variables that may appear causally related but are not. These spurious correlations introduce undesired bias to the learning problem, posing a significant challenge in various machine learning applications. Here are a few illustrative examples of this phenomenon.

- In recommendation systems, user behavior and preferences are often influenced by conformity, which refers to the tendency of individuals to align with larger groups or social norms. Users may feel inclined to conform to popular trends or recommendations. Ignoring the impact of conformity can lead to an overestimation of a user's preference for certain items (Gao et al., 2022);

- In image classification, when dogs frequently appear alongside grass in the training set, the classifier may incorrectly label an image of grass as a dog. This misclassification arises because the model relies on the background (the irrelevant factors) rather than focusing on the specific pixels that correspond to dogs (the actual causal) (Zhang et al., 2021a; Wang et al., 2021c).

- When determining the ranking of tweets, the use of gender icons in tweets is usually not causally related to the number of likes; their statistical correlation comes from the topic, as it influences both the choice of icon and the audience. Therefore, it is not appropriate to determine the ranking by gender icons (Feder et al., 2022).

If we want to apply RL in real-world scenarios, it is important to be mindful of spurious correlations, especially when the agent is working with biased data. For instance, when optimizing long-term user satisfaction in multiple-round recommendations, there is often a spurious correlation between exposure and clicks in adjacent timesteps. This is because they are both influenced by item popularity. From another perspective, when we observe a click, it may depend on user preference or item popularity, which creates a spurious correlation between the two factors. In both scenarios, agents may make incorrect predictions or decisions, such as only recommending popular items (a suboptimal policy for both the system and the user), and this can further cause filter bubbles. In a nutshell, if the agent learns a spurious correlation between two variables, it may mistakenly believe that changing one will affect the other. This misunderstanding can lead to suboptimal or even harmful behavior in real-world decision-making problems.

## 4.2 How can Causality Help to Address Spurious Correlations?

The non-causal approaches lack a language for systematically discussing spurious correlations. From the causal perspective, spurious correlations arise when the data generation process involves unobserved confounders (common cause) or when a collider node (common effect) serves as the condition. The former leads to confounding bias, while the latter results in selection bias. See Figure 6 for a visual interpretation of these phenomena. Causal graphs enable us to trace the source of spurious correlations by closely scrutinizing the data generation process. To eliminate the bias induced by spurious correlations, it is necessary to make decisions regarding causality instead of statistical correlations. This is where causal reasoning comes in: It provides principled tools to analyze and deal with confounding and selective bias (Pearl, 2009b; Bareinboim et al., 2014; Glymour et al., 2016; Bareinboim et al., 2022), helping RL agents accurately estimate the causal effects in decision-making problems.

Table 2: Selected methods utilizing causality to address spurious correlations.

| Category | Paper | Techniques | Settings | Environments or Tasks |
|---|---|---|---|---|
| Confounding bias | Zhang et al. (2020b) | Causal graph [5] | imitation | Toy |
| | Kumor et al. (2021) | Causal graph | imitation | Toy |
| | Swamy et al. (2022) | Instrumental variables | imitation | Lunar Lander (OpenAI Gym) |
| | | | | Locomotion (PyBullet Gym) |
| | Namkoong et al. (2020) | Sensitivity analysis | offline | Toy |
| | Bennett et al. (2021) | Proxy variables | offline | Toy |
| | Lu & Lobato (2018) | Backdoor adjustment | offline | Pendulum (OpenAI Gym) |
| | | | | Cart Pole (OpenAI Gym) |
| | | | | MNIST |
| | Zhang & Bareinboim (2019) | Causal bound | online* | Toy |
| | | Sensitivity analysis | | |
| | Rezende et al. (2020) | Backdoor adjustment | offline | Toy |
| | | | | MiniPacman |
| | | | | 3D Maze (Unity) |
| | Zhang & Bareinboim (2020) | Causal graph | online* | Toy |
| | | Causal bound | | |
| | Wang et al. (2021b) | Frontdoor adjustment | online* | - |
| | | Backdoor adjustment | | |
| | Liao et al. (2021) | Instrumental variables | offline | - |
| | Guo et al. (2022a) | Instrumental variables | offline | - |
| | Gasse et al. (2023) | Backdoor adjustment | online* | Toy |
| | Pace et al. (2023) | Causal graph | offline | Sepsis |
| | | | | HiRID |
| | Yang et al. (2022a) | Causal graph | online | Toy |
| | | | | Cart Pole (OpenAI Gym) |
| | | | | Lunar Lander (OpenAI Gym) |
| | Zhang & Bareinboim (2022b) | Causal graph | online* | Toy |
| Selection bias | Bai et al. (2021) | Inverse probability weighting | online | Manipulation (OpenAI) |
| | Deng et al. (2021) | Causal graph | offline | D4RL |

One may assume that on-policy RL is immune to spurious correlations since it directly learns causal effects of the form $p(r|s, \text{do}(a))$ from interventional data. However, it is important to note that understanding the effect of an action on the outcome alone is insufficient for comprehending the complete data generation process. The influence of different covariates on the outcome also plays a crucial role. For example, in personalized recommendation systems, the covariates can be divided into relevant and spurious features. If the training environment consistently pairs the clicked items with specific values of spurious features (e.g., clickbait in textual features), the agent may unintentionally learn a policy based on these spurious features. When the feature distribution changes in the test environment, the agent may make erroneous decisions (Gao et al., 2022). This example demonstrates the prevalence of spurious correlations in real-world decision-making problems. Demystifying the causal relationships helps resolve such challenges.

### 4.3 Causal Reinforcement Learning for Addressing Spurious Correlations

Based on the underlying causal structure of the decision-making problem, spurious correlations can manifest in two distinct types: confounding bias arising from fork structures, and selective bias arising from collider structures (as depicted by Figure 6). Accordingly, we categorize existing methods into two groups. Additionally, to be self-contained, we incorporate studies on imitation learning (IL) and off-policy evaluation (OPE), given their close relevance to policy learning in RL. The representative works are shown in Table 2.

#### 4.3.1 Addressing Confounding Bias

We start by introducing an important technique in causal inference named do-calculus (Pearl, 1995). It is an axiomatic system that enables us to replace probability formulas containing the do operator with ordinary conditional probabilities. The do-calculus includes three axiom schemas that provide graphical criteria

---

[5]Causal graph as a technique refers to using causal graphs to describe the data generation process, and designing graphical criteria for determining properties such as identifiability or developing algorithms based on causal graphs.

for making certain substitutions. It has been proven to be complete for identifying causal effects (Huang & Valtorta, 2006; Shpitser & Pearl, 2006). Derived from the do-calculus, the backdoor and frontdoor adjustment are two widely used methods for eliminating confounding bias Glymour et al. (2016); Bareinboim et al. (2022). The key intuition is to block spurious correlations between the treatment and outcome variables that pass through the confounders. In situations where unobserved confounding exists, it is still possible to identify the causal effect of interest if observed proxy variables for the confounder are available (Miao et al., 2018; Ghassami et al., 2023). Another popular approach for addressing unobserved confounding is to identify and employ instrumental variables (Angrist et al., 1996; Baiocchi et al., 2014). An instrumental variable, denoted as $I$, must satisfy three conditions: 1) $I$ is a cause of $T$ (the treatment variable); 2) $I$ affects $Y$ (the outcome variable) only through $T$; and 3) The effect of $I$ on $Y$ is unconfounded. Since the instrumental variable $I$ influences $Y$ only through $T$, and this effect is unconfounded, we can indirectly estimate the effect of $T$ on $Y$ through the effect of the instrumental variable $Z$ on $Y$. Furthermore, we can evaluate the robustness of the estimated causal effect against unobserved confounding by varying the strength of the confounder's effect, which is referred to as sensitivity analysis (Díaz & van der Laan, 2013; Kuang et al., 2020).

Zhang et al. (2020b) studied a single-step imitation learning problem using a combination of demonstration data and structural knowledge about the data-generating process. They proposed a graphical criterion for determining the feasibility of imitation learning in the presence of unobserved confounders and a practical procedure for estimating valid imitating policies with confounded expert data. This approach was then extended to the sequential setting in a subsequent paper (Kumor et al., 2021). Swamy et al. (2022) designed an algorithm for imitation learning with corrupted data. They proposed to use instrumental variable regression (Stock & Trebbi, 2003) to resolve the spurious correlations caused by unobserved confounding.

Several research focused on the OPE problem, which seeks to estimate the performance of policies using data generated by different policies. For example, Namkoong et al. (2020) conducted sensitivity analyses on OPE methods under unobserved confounding. They derived worst-case bounds on the performance of an evaluation policy and proposed an efficient procedure for estimating these bounds with statistical consistency, allowing for a reliable selection of policies. Bennett et al. (2021), on the other hand, proposed a new estimator for OPE in infinite-horizon RL. The authors established the identifiability of the policy value from off-policy data by employing a latent variable model for states and actions (which can be seen as proxy variables for the unobserved confounders). The authors further presented a strategy for estimating the stationary distribution ratio using proxies, which is then utilized for policy evaluation.

Lu & Lobato (2018) introduced a novel approach called deconfounding RL, aiming to learn effective policies from historical data affected by unobserved factors. Their method begins by estimating a latent-variable model using observational data, which identifies latent confounders and assesses their impact on actions and rewards. Then these confounders are utilized in backdoor adjustment to address confounding bias, enabling the policy to be optimized based on a deconfounding model. Experimental results demonstrate the method's superiority over traditional RL approaches when applied to observational data with confounders. Liao et al. (2021) also focused on the offline setting. They found that RL practitioners often encounter unobserved confounding in medical scenarios, but there are some common sources of instrumental variables, including preferences, distance to specialty care providers, and genetic variants (Baiocchi et al., 2014). Therefore, they proposed an efficient algorithm to recover the transition dynamics from observational data using instrumental variables, and employ a planning algorithm, such as value iteration, to search for the optimal policy. Similarly, Guo et al. (2022a) embraced a comparable approach when tackling the POMDP problem. They addressed the estimation of the transfer kernel by framing it as a series of confounded regression problems that can be solved by selecting suitable instrumental variables. After constructing confidence regions for the model parameters, the final policy can be derived using a pessimistic planning method within these regions. Wang et al. (2021b), on the other hand, studied how confounded observational data can facilitate exploration during online learning. They proposed the deconfounded optimistic value iteration algorithm, which combines observational and interventional data to update the value function estimates. This algorithm effectively handles confounding bias by leveraging backdoor and frontdoor adjustment, achieving a smaller regret than the optimal online-only algorithm. Gasse et al. (2023) studied MBRL for POMDP problems. Specifically, They used ideas from the do-calculus framework to formulate model learning as a causal inference problem. They introduced a novel method to learn a latent-based causal transition model capable of

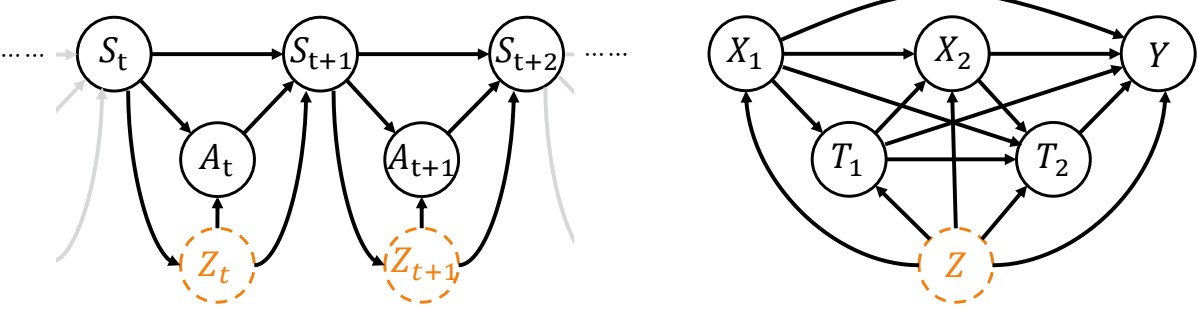

Figure 7: Causal graphs illustrating the confounded MDP (left) discussed in Wang et al. (2021b) and a DTR with two stages of treatments (right). The relationship between these two settings can be derived easily. If none of the unobserved confounders $\{Z_t\}$ are influenced by the states $\{S_t\}$, we can aggregate them into a global confounder $Z$. Let $Y$ denotes the outcome of treatments in a DTR, and let $X$ and $T$ represent the covariates and treatments, respectively. By decomposing the states into covariates and treatments at each step (e.g., $s_2 = \{X_1, T_1, X_2\}$), a confounded MDP reduces to a DTR.

explaining both interventional and observational regimes. By utilizing the latent variable, they were able to recover the standard transition model, which, in turn, facilitated the training of RL agents using simulated transitions. While the majority of existing research focuses on addressing identifiable hidden confounding, such as using instrumental or backdoor variables, hidden confounding is possibly non-identifiable in some real-world scenarios. Pace et al. (2023) showed that even in such scenarios, a careful analysis from a causal perspective can still contribute to significant improvements in policy learning. Specifically, they proposed a new type of uncertainty known as "delphic uncertainty", which, in contrast to the widely studied epistemic uncertainty and aleatoric uncertainty, quantifies the variability across world models compatible with the observational data. They devised an offline RL algorithm that incorporates this uncertainty as a Bellman penalty. Their algorithm demonstrates effectiveness in reducing non-identifiable confounding bias on two medical datasets. Yang et al. (2022a) proposed the causal inference Q-network algorithm to address confounding bias arising from various types of observational interferences, such as Gaussian noise and observation black-out. The authors began by analyzing the impact of these interferences on the decision-making process using causal graphs. They then devised a novel algorithm that incorporates the interference label to learn the relationship between interfered observations and Q-values in order to maximize rewards in the presence of observational interference. The model learns to infer the interference label and adjusts the policy accordingly. Experimental results demonstrate the algorithm's improved resilience and robustness against different types of interferences. Rezende et al. (2020) discussed the application of partial models in RL, a model-based approach that does not require modeling the complete (and usually high-dimensional) observation. They showed that the causal correctness of a partial model can be influenced by behavior policies, leading to an overestimation of the rewards associated with suboptimal actions. To address this issue, the authors proposed a simple yet effective solution. Since we have full control of the agent's computational graph, we can choose any node situated between the internal state and the produced action as a backdoor variable, e.g., the intended action. By applying backdoor adjustment, we can ensure the causal correctness of partial models.

A less familiar but highly important research topic for RL researchers is dynamic treatment regimes (DTRs) (Murphy, 2003). Closely related to the fields of biostatistics, epidemiology, and clinical medicine, this topic focuses on determining personalized treatment strategies, including dosing or treatment planning. It aims to maximize the long-term clinical outcome and can be mathematically modeled as an MDP with a global confounder, as depicted in Figure 7. In real-world medical scenarios, global confounders such as genetic characteristics and lifestyle often exist but may not be observed or recorded due to various reasons. When applying RL to learn DTRs, we must properly handle the confounders; thus, it is highly relevant to the topic discussed in this section.

Zhang & Bareinboim (2019) considered a setting in which causal effect is not identifiable. They proposed an online learning algorithm to solve DTRs by combining confounded observational data. Their algorithm hinges on sensitivity analyses and incorporates causal bounds to accelerate the learning process. This online learning setting has gained popularity as it allows for conducting sequential and adaptive experimentation to maximize the outcome variable. However, a significant challenge arises when dealing with a vast space of covariates and treatments, where online learning algorithms may result in unacceptable regret. To address this challenge, Zhang & Bareinboim (2020) proposed an efficient procedure that leverages the structural knowledge encoded in the causal graph to reduce the dimensionality of the candidate policy space. By exploiting the sparsity of the graph, where certain covariates are affected by only a small subset of treatments, the proposed method exponentially reduces the dimensionality of the learning problem. The experimental results consistently demonstrate that the proposed method outperforms state-of-the-art (SOTA) methods in terms of performance. More recently, Zhang & Bareinboim (2022b) further explored the challenge of policy space with mixed scopes, i.e., the agent has to optimize over candidate policies with varying state-action spaces. This becomes particularly crucial in medical domains such as cancer, HIV, and depression, where finding the optimal combination of treatments is essential for achieving desired outcomes. To tackle this issue, the authors propose a novel method that utilizes causal graphs to identify the maximal sets of variables that are causally related to each other. These sets are then utilized to parameterize SCMs, enabling the representation of different interventional distributions using a minimal set of components, thereby enhancing the efficiency of the learning process.

A notable trend that emerges from the above literature is the combination of observational and experimental data from diverse sources, commonly referred to as data fusion (Bareinboim et al., 2022). This idea has been proven to be effective in addressing complex problems related to reasoning and decision-making (Zhang & Bareinboim, 2019; Lee et al., 2020b; Correa et al., 2021; Zhang et al., 2022b; Gasse et al., 2023). It represents an important research direction in causal inference and may have profound implications for reinforcement learning. In reinforcement learning, agents often have access to both observational and experimental data, e.g., augmenting online learning with confounded observational data collected by other policies. As a result, data fusion introduces new challenges and holds promising potential for the research community.

### 4.3.2   Addressing Selection Bias

Selective bias occurs when data samples fail to represent the target population. For example, selective bias arises when researchers seek to understand the effect of a certain drug on curing a disease by investigating patients in a selected hospital. This is because those patients may differ significantly from the population regarding their residence, wealth, and social status, making them unrepresentative.

Bai et al. (2021) investigated the selective bias associated with using hindsight experience replay (HER) in goal-conditioned reinforcement learning (GCRL) problems. In particular, HER relabels the goal of each collected trajectory and computes new rewards, thereby allowing the agent to learn from failure experiences —— instances where it failed to reach the original goal but succeeded in reaching the relabeled one. However, the relabeled target distribution may not accurately represent the original target distribution. This discrepancy can misguide agents trained with HER, leading them to mistakenly believe that repeating decisions made in the failure experience will result in high rewards. To address this issue, the authors proposed to use inverse probability weighting, a technique in causal inference, to assign appropriate weights to the rewards computed for the relabeled goals. By reweighting the samples, the agent can mitigate the selection bias induced by HER and effectively learn from a balanced mixture of successful and unsuccessful outcomes, ultimately enhancing the overall performance. Deng et al. (2021) examined the offline RL problem through the lens of selective bias. In the offline setting, agents are vulnerable to the spurious correlation between uncertainty and decision-making, which can result in learning suboptimal policies. Taking a causal perspective, the empirical return is the outcome of both uncertainty and actual return. Since it is infeasible to reduce uncertainty by acquiring more data in the offline setting, an agent might mistakenly assume a causal relationship between uncertainty and actual return. As a result, it may favor policies that achieve high returns by chance (high uncertainty). To address this issue, the authors propose quantifying uncertainty and using it as a penalty term in the learning process. The results show that this method outperforms various baselines that do not consider spurious correlations in the offline setting.

In general, a standard reinforcement learning setup allows agents to learn from the same environment on which they will be tested, so issues related to sample representativeness are often not a primary concern. However, as discussed above, specific algorithmic choices or offline learning can introduce selection bias. Overall, this issue remains relatively underexplored in reinforcement learning. In the realm of causal inference, substantial efforts have been dedicated to addressing the challenge of selection bias. For instance, the graphical conditions proposed by Bareinboim et al. (2014) offer a method for assessing the recoverability of certain conditional probabilities and causal effects from data affected by selection bias. These efforts provide valuable tools for further investigations into this issue in reinforcement learning.

In summary, this section presents some instances of spurious correlations in reinforcement learning and the causal RL methods to solve these problems. A careful reader may notice the subtle connection between spurious correlation and generalization. For instance, we can treat unobserved confounders as contextual variables and create new domains by applying external interventions to these variables. In such cases, policies that exhibit strong generalization (robust to different contexts) tend to be less sensitive to spurious correlations, and vice versa. Therefore, the distributional robustness framework designed for generalization may also serve as a tool to mitigate spurious correlations under certain conditions (Ding et al., 2023). However, by comparison, causal RL methods for addressing generalization and spurious correlation may not share the same focus and techniques. The former is usually performance-oriented, focusing on how well a learned policy performs on new domains. In contrast, the latter is more focused on identifying the causal effects of interest rather than dealing with the distributional shifts associated with confounding variables. Technically, research on spurious correlations often introduces structural assumptions (e.g., causal graphs) and employs causal inference techniques like backdoor/frontdoor adjustments to eliminate biases. In recent years, there has been an interesting trend (Zhang et al., 2021a; Cui & Athey, 2022; Ding et al., 2023) in exploring the intersection of the two research fields, which has the potential to offer valuable insights and techniques to researchers in both fields.

## 5 Enhancing Sample Efficiency through Causal Reinforcement Learning

### 5.1 The Issue of Sample Efficiency in Reinforcement Learning

In RL, training data is typically not provided before interacting with the environment. Unlike supervised and unsupervised learning methods that directly learn from a fixed dataset, an RL agent needs to actively gather data to optimize its policy towards achieving the highest return. An effective RL algorithm should be able to master the optimal policy with as few experiences as possible (in other words, it need to be sample-efficient). Current methods often require collecting millions of samples to succeed in even simple tasks, let alone more complicated environments and reward mechanisms. For example, AlphaGo Zero was trained over roughly $3 \times 10^7$ games of self-play (Silver et al., 2017); OpenAI's Rubik's Cube robot took nearly $10^4$ years of simulation experience (OpenAI et al., 2019). This inefficiency entails a high training cost and prevents the use of RL techniques for solving real-world decision-making problems. Therefore, the sample efficiency issue is a core challenge in RL, necessitating the development of RL algorithms that can save time and computational resources. In this section, in addition to examining sample-efficient algorithms that are designed to minimize regret in online learning, we also cover a number of approaches related to knowledge transfer. These methods require only a small number of samples from the target domain to converge after pre-training on the source domains, thus indirectly enhancing the sample efficiency within the target domain.

### 5.2 Causal Reinforcement Learning for Addressing Sample Inefficienty

Sample efficiency has been extensively explored in RL literature (Kakade, 2003; Osband et al., 2013; Grande et al., 2014; Yu, 2018) and causality offers some valuable principles for designing sample-efficient RL algorithms. Accordingly, we can organize existing research into three main lines: representation learning, directed exploration, and data augmentation. The representative works are shown in Table 3.

Table 3: Selected methods utilizing causality to optimize sample efficiency.

| Category | Paper | Techniques | Settings | Environments or Tasks |
|---|---|---|---|---|
| Representation Learning | Sontakke et al. (2021) | Causal representation learning | offline to online | Manipulation (CausalWorld) |
| | Lee et al. (2021) | Intervention Domain randomization | online | Manipulation (Isaac Gym) |
| | Huang et al. (2022b) | Causal dynamics learning | online* | Car Racing (OpenAI Gym) VizDoom |
| | Wang et al. (2022) | Causal dynamics learning | online* | Chemical Manipulation (robosuite) |
| Directed Exploration | Seitzer et al. (2021) | Intervention | online | Manipulation (OpenAI) |
| Data Augmentation | Buesing et al. (2019) | Counterfactual reasoning | online | Sokoban |
| | Lu et al. (2020) | Counterfactual reasoning | online | Cart Pole (OpenAI Gym) MIMIC-III |
| | Pitis et al. (2020) | Counterfactual reasoning | offline | Spriteworld Pong (Roboschool) Manipulation (OpenAI) |
| | Zhu et al. (2021) | Counterfactual reasoning | online | Manipulation (CausalWorld) |

### 5.2.1 Representation Learning for Sample Efficiency

A good representation of the environment can be beneficial for sample-efficient RL. By providing a compact and informative representation of the environment, an RL agent can learn more effectively with fewer samples. This is because a good representation can help the agent identify important features of the environment and abstract away unnecessary details, allowing the agent to make better use of its experiences.

Motivated by the principle of independent causal mechanisms (Schölkopf et al., 2021), Sontakke et al. (2021) argue that the information in an observed trajectory is the sum of information "injected" by different causes. Thus, when learning in multiple environments with different physical properties, if the collected trajectories are well-clustered, it suggests that there may be only a single causal feature that differs among these trajectories, such as the mass, size, or shape of an object. Based on this idea, they employ clustering performance to induce experimental behavior and use clustering results as a surrogate for causal representation. With the state augmented by these representations, the learned policies exhibit outstanding zero-shot generalization ability and require only a small number of training samples to converge in new environments.

Another way to improve sample efficiency through representations is state abstraction. Lee et al. (2021) assumed the availability of an environmental model and used causal reasoning to identify relevant context variables. Specifically, the proposed method conducts interventions to alter one context variable at a time and observes the causal influence on the outcome. This approach effectively reduces the dimensionality of the state space and simplifies the learning problem. However, in many scenarios, direct intervention may not be feasible. Some non-causal approaches (Jong & Stone, 2005; Zhang et al., 2022a) achieve abstraction by aggregating states that yield the same reward sequence. While these

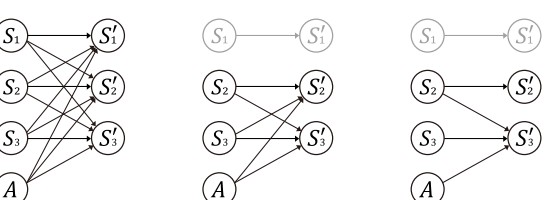

Figure 8: An illustration of a state transition between adjacent time steps. (Left) No abstraction. All variables are fully connected. (Middle) An irrelevant covariate $S_1$ is removed but the rest are still fully connected. (Right) Only the causal edges are preserved.

approaches help reduce the dimensionality of the state space, they still suffer from redundant dependencies and are vulnerable to spurious correlations. In contrast, causal relationships in the real world are typically sparse and stable(Schölkopf et al., 2021; Huang et al., 2022a), leading to more effective abstraction. See Figure 8 for a comparison between these two types of abstraction.

When observations involve high-dimensional and low-level data, causal representation learning helps identify the relevant high-level concepts for the given tasks. Therefore, many causal RL approaches are motivated by the idea that exploiting the true causal structure of the problem will reduce the complexity of learning, and thus improve sample efficiency. For example, Huang et al. (2022b) introduced a method to learn action-

sufficient state representations from data collected by a random policy. These representations consist of a minimal set of state variables that contain sufficient information for decision-making. Wang et al. (2022), on the other hand, studied task-independent state abstraction which only omits action-irrelevant variables that neither change with actions nor influence actions' results, identified through independence tests based on conditional mutual information. Their approach includes variables that are potentially useful for future tasks rather than being restricted to a particular training task.

### 5.2.2 Directed Exploration for Sample Efficiency

While a good representation of the environment is beneficial, it is not necessarily sufficient for sample efficient RL (Du et al., 2020). To improve sample efficiency, researchers have been studying various exploration strategies (Yang et al., 2022b). Some research has drawn inspiration from developmental psychology (Ryan & Deci, 2000; Barto, 2013) and used intrinsic motivation to motivate agents to explore unknown environments efficiently (Pathak et al., 2017; Burda et al., 2022). This can be done by giving bonuses to exploratory behaviors that discover novel or uncertain states. However, from a causal perspective, it is important to note that not all regions of high uncertainty are equally important. Those regions which establish a causal relationship with the task's success are more worthy of exploration.

Seitzer et al. (2021) studied the problem of directed exploration in robotic manipulation tasks, where the agent must physically interact with the target object to generate valuable data before acquiring complex manipulation skills such as object relocation. The authors proposed a method to quantify the causal influence of actions on the object and incorporated it into the exploration process to guide the agent. Experimental results demonstrate that the proposed method significantly improves the sample efficiency across various robotic manipulation tasks. On the other hand, Sontakke et al. (2021) introduced a method to learn self-supervised experiments based on the principle of independent causal mechanisms. This method utilizes the concept of One-Factor-at-A-Time (OFAT), wherein good experimental behavior should examine one factor at a time while keeping others constant. The rationale behind this approach is that altering only one causal factor should yield less information compared to changing multiple factors simultaneously. Consequently, the learning problem of experimental behavior can be reformulated as minimizing the amount of information contained in the generated data (also referred to as maximizing "causal curiosity"). Empirical results show that RL agents pre-trained with causal curiosity exhibit improved efficiency in solving new tasks.

### 5.2.3 Data Augmentation for Sample Efficiency

Data augmentation is a common machine learning technique aimed at improving algorithm performance by generating additional training data. Counterfactual data augmentation is a causality-based approach that uses a causal model to imitate the environment and generate data that is unobserved in the real world. This is particularly useful for RL problems because collecting large amounts of real-world data is often difficult or expensive. By simulating diverse counterfactual scenarios, RL agents can determine the effects of different actions without interacting with the environment, resulting in sample-efficient learning.

The implementation of counterfactual data augmentation follows a counterfactual reasoning procedure [6] that consists of three steps (Pearl, 2009a, Chapter 7), as demonstrated in Figure 9:

1. **Abduction** is about using observed data to infer the values of the exogenous variables $\mathcal{U}$;

2. **Action** involves modifying the structural equations of the variables of interest in the SCM; and

3. **Prediction** uses the modified SCM to generate counterfactual data by plugging the exogenous variables back into the equations for computation.

---

[6]This procedure intuitively elucidates the computational process for evaluating counterfactual statements based on available evidence. Nonetheless, it is not always feasible in practice. Abduction necessitates access to the distribution of exogenous variables, while action and prediction require knowledge about the underlying functional relationships. In the causal inference literature, extensive research has been dedicated to investigating the combination of qualitative assumptions and data to identify counterfactual distributions. For more details, please refer to Shpitser & Pearl (2007); Correa et al. (2021); Zhang et al. (2022b).

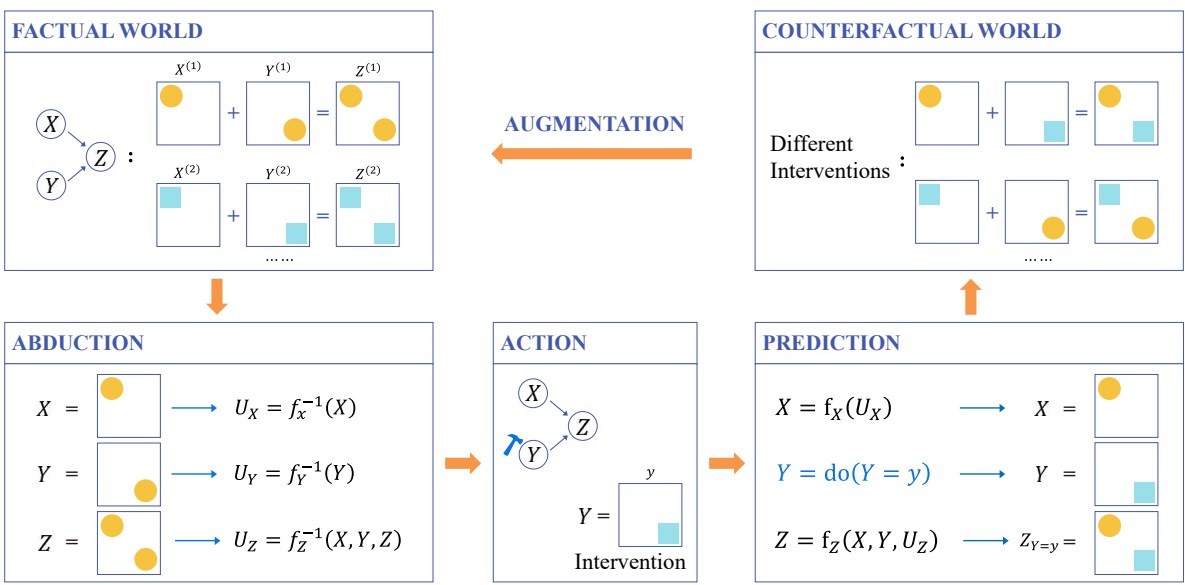

Figure 9: An example of counterfactual data augmentation following the counterfactual reasoning procedure: abduction, action, and prediction. The outcome of this procedure is then used to augment the training data observed in the factual world.

While traditional model-based reinforcement learning (MBRL) methods (Wang et al., 2019; Luo et al., 2022) can also generate samples by fitting a probabilistic density model, they lack the capability to effectively model exogenous variables. This limitation can lead to under-fitting when dealing with complex distributions of exogenous variables (Buesing et al., 2019). In contrast, counterfactual data augmentation explicitly incorporates exogenous variables using the SCM framework. From a Bayesian perspective, traditional MBRL approaches implicitly rely on a fixed prior, such as the Gaussian distribution, for exogenous variables, whereas counterfactual data augmentation leverages additional information (evidence) from the collected data to estimate the posterior distribution of exogenous variables. Consequently, counterfactual data generation holds promise in producing high-quality training data, potentially leading to improved policy evaluation and optimization.

Buesing et al. (2019) proposed the counterfactually-guided policy search (CF-GPS) algorithm, designed to search for the optimal policies in POMDPs. They framed model-based POMDP problems using SCMs. The proposed algorithm evaluates the outcome of counterfactual actions based on real experience, thereby improving the utilization of experience data. In a similar vein, Lu et al. (2020) proposed a sample-efficient RL algorithm based on the SCM framework. Their objective was to address issues related to mechanism heterogeneity and data scarcity. Their approach empowers agents to evaluate the potential consequences of counterfactual actions, thereby circumventing the need for actual exploration and alleviating biases arising from limited experience. Pitis et al. (2020) presented a novel framework that leverages a locally factored dynamics model to generate counterfactual transitions for RL agents. Specifically, the term "locally factored" indicates that the state-action space can be partitioned into a disjoint union of local subsets, each has its own causal structure. This locally factored approach allows for an exponential reduction in the sample complexity of training a dynamics model and enables reliable generalization to unseen states and actions. More recently, Zhu et al. (2021) proposed a novel approach to overcome the limitations of existing MBRL methods in the context of robotic manipulation tasks. These tasks are particularly challenging due to the diversity of object properties and the risk of robot damage. The proposed method uses SCMs to capture the underlying environmental dynamics and generate counterfactual episodes involving rarely seen or unseen objects. Experimental results demonstrate superior sample efficiency, requiring fewer environment steps to converge compared to existing MBRL algorithms.

Table 4: Selected methods utilizing causality for goals beyond maximizing returns.

| Category | Paper | Techniques | Settings | Environments or Tasks |
|---|---|---|---|---|
| Explainability | Foerster et al. (2018) | Counterfactual | online | StarCraft |
| | Madumal et al. (2020) | Counterfactual reasoning | online | OpenAI Gym
StarCraft |
| | Bica et al. (2021a) | Counterfactual reasoning | imitation | Toy
MIMIC-III |
| | Mesnard et al. (2021) | Counterfactual reasoning | online | Toy
Key-to-Door (Not accessible)
Interleaving (Not accessible) |
| | Tsirtsis et al. (2021) | Counterfactual reasoning | online | Toy
Therapy |
| | Triantafyllou et al. (2022) | Counterfactual reasoning | online | Goofspiel (Not accessible) |
| | Herlau & Larsen (2022) | Mediation analysis | online | Toy
DoorKey (Not accessible) |
| Fairness | Zhang & Bareinboim (2018) | Counterfactual reasoning
Mediation analysis | offline | Toy |
| | Huang et al. (2022c) | Causal graph
Causal reasoning | online | Toy |
| | Balakrishnan et al. (2022) | Causal graph
Counterfactual reasoning | online | Toy |
| Safety | Hart & Knoll (2020) | Counterfactual reasoning | offline | BARK-ML |
| | Everitt et al. (2021) | Causal graph | online | - |

# 6 Promoting Explainability, Fairness, and Safety with Causal Reinforcement Learning

In general, the primary objective of RL is to maximize returns. However, with the increasing integration of RL-based automated decision systems into our daily lives, it becomes imperative to examine the interactions between RL agents and humans, as well as their potential societal implications. In this section, we explore causal RL methods that aim to address and alleviate challenges related to explainability, fairness, and safety. The representative works are shown in Table 4.

## 6.1 Explainability

### 6.1.1 Explainability in Reinforcement Learning

Explainability in RL refers to the ability to understand and interpret the decisions made by an RL agent. It is important to both researchers and general users. Explanations reflect the knowledge learned by the agent, facilitating in-depth understanding. They also allow researchers to participate efficiently in the design and continual optimization of an algorithm. Furthermore, explanations unveil the internal logic of the decision-making process. When agents outperform humans, we can extract valuable insights from these explanations to inform human practice within a specific domain. For general users, explanations provide a rationale behind each decision, thereby enhancing their comprehension of intelligent agents and instilling greater confidence in the agent's capabilities.

### 6.1.2 Explainable Reinforcement Learning with Causality

Explainable RL methods can be broadly categorized into two groups: post hoc and intrinsic approaches (Puiutta & Veith, 2020; Heuillet et al., 2021). Post hoc explanations are provided after the model execution, whereas intrinsic approaches inherently possess transparency. Post hoc explanations, such as the saliency map approach (Greydanus et al., 2018; Mott et al., 2019), often rely on correlations. However, as we mentioned earlier, conclusions drawn based on correlations may be unreliable and fail to answer causal questions. On the other hand, intrinsic explanations can be achieved using interpretable algorithms,

such as linear regression or decision trees, but the limited modeling capacity of these algorithms may prove insufficient in explaining complex behaviors (Puiutta & Veith, 2020).

In contrast, humans possess an innate and powerful ability to explain the connections between different events through a "mental causal model (Sloman, 2005)". This cognitive ability enables us to employ causal language in our everyday interactions, using phrases such as "because," "therefore," and "if only." By harnessing causal relationships, we acquire natural and flexible explanations that do not rely on specific algorithms or models, thereby greatly facilitating efficient communication and collaboration. Drawing inspiration from human cognition, we can integrate causality into RL to provide explanations that enable agents to articulate their decisions and comprehension of the environment and tasks using causal language. Furthermore, in cases when the agent makes mistakes, we can respond with tailored solutions guided by causal insights.

One way to generate explanations is to use the concept of counterfactuals such as exploring the minimal changes necessary to produce a different outcome. The term "counterfactual" is popular in multi-agent reinforcement learning (MARL). For example, Foerster et al. (2018) proposed a method named counterfactual multi-agent policy gradients for efficiently learning decentralized policies in cooperative multi-agent systems. More precisely, counterfactuals help resolve the challenge of multi-agent credit assignment so that agents and humans can better understand the contribution of individual behavior to the team. Some subsequent studies followed the same idea (Su et al., 2020; Zhou et al., 2022). These approaches did not perform the complete counterfactual reasoning procedure as shown in Figure 9, missing the critical step of abduction, which offers opportunities for further enhancements. More recently, Triantafyllou et al. (2022) established a connection between Dec-POMDPs and SCM, enabling them to investigate the credit assignment problem in MARL using causal language. They proposed to formalize the notion of responsibility attribution based on actual causality, as defined by counterfactuals, which is a significant stride in developing a rigorous framework that supports accountable MARL research. Mesnard et al. (2021), on the other hand, studied the temporal credit assignment problem, i.e., measuring an action's influence on future rewards. Inspired by the concept of counterfactuals from causality theory, the authors proposed conditioning value functions on future events, which separate the influence of actions on future rewards from the effects of other sources of stochasticity. This approach not only facilitates explainable credit assignment but also reduces the variance of policy gradient estimates.

Madumal et al. (2020) used theories from cognitive science to explain how humans understand the world through causal relationships and how these relationships can help us understand and explain the behavior of RL agents. They presented an approach that integrates an SCM into reinforcement learning and used the learned model to generate explanations of behavior based on counterfactual analysis. For example, when quiring why a Starcraft II agent builds supply depots instead of barracks (a typical counterfactual query), the agent can respond by explaining that constructing supply depots is more desirable, as it helps increase the number of destroyed units and buildings. To evaluate the proposed approach, a study was conducted involving 120 participants. The results demonstrate that the causality-based explanations outperformed other explanation models in terms of understanding, explanation satisfaction, and trust. Bica et al. (2021a) proposed an innovative approach to gain insights into expert decision processes by integrating counterfactual reasoning into batch inverse reinforcement learning. Their method focuses on learning explanations of expert decisions by modeling their reward function based on preferences with respect to counterfactual outcomes. This framework is particularly helpful in real-world scenarios where active experimentation may not be feasible. Tsirtsis et al. (2021) conducted a study on the identification of optimal counterfactual explanations [7] for a sequential decision process. They approached this problem by formulating it as a constrained search problem and devised a polynomial time algorithm based on dynamic programming to find the solution. Specifically, this problem requires the algorithm to use a causal model of the environment to search for another sequence of actions that differs from the observed sequence of actions by a specified number of actions. The study conducted by Herlau & Larsen (2022) explores the application of mediation analysis in RL. The proposed method focuses on training an RL agent to optimize natural indirect effects, which allows for identifying critical decision points. For instance, in the task of unlocking a door, the agent can effectively

---

[7]Remark that the term "counterfactual explanation" is commonly used in the field of explainability. It refers to a broad range of methods that offer explanations by analyzing the changes that could result from altering the inputs to a model in a particular way. These methods do not necessarily imply causality. For further details, please refer to Verma et al. (2022).

recognize the event of acquiring a key. By leveraging mediation analysis, the agent can acquire a concise and interpretable causal model, enhancing its overall performance and explanatory capabilities.

## 6.2 Fairness

### 6.2.1 Fairness in Reinforcement Learning

In addition to explainability, we also want agents to align with human values and avoid potential harm to human society, with fairness being a key consideration. As machine learning applications continue to permeate various aspects of our daily lives, fairness is increasingly recognized as a significant concern by business owners, general users, and policymakers (Carey & Wu, 2022). In real-world scenarios, fairness concerns often exhibit a dynamic nature (Gajane et al., 2022), involving multiple decision rounds. For example, resource allocation and college admissions can be modeled as MDPs (D'Amour et al., 2020), wherein actions have cumulative effects on the population, leading to dynamic changes in fairness. Ignoring this dynamic nature of a system may lead to unintended consequences, e.g., exacerbating the disparity between advantaged and disadvantaged groups (Liu et al., 2018; Creager et al., 2020; D'Amour et al., 2020). In decision-making problems like these, RL agents should strive to genuinely benefit humans and promote social good, avoiding any form of discrimination or harm towards specific individuals or groups.

### 6.2.2 Fair Reinforcement Learning with Causality

When considering the application of reinforcement learning to solve fairness-aware decision-making problems, it is crucial to first examine the available prior knowledge. Detailed causal modeling allows us to characterize and understand the intricate interplay between decision-making and environmental dynamics (Zhang et al., 2020c; Tang et al., 2023). Specifically, we can determine what observable variables and latent factors are involved in a specific problem and how they affect (long-term) fairness. For example, Balakrishnan et al. (2022) used causal graphs to study fairness principles in decision-making problems. They encoded these principles into causal, non-causal and utility components, and analyzed the relationship between different causal paths and fairness. The author then turned fairness measures into constraints to enforce certain fairness principles, thereby framing fair policy learning into a constrained optimization problem.

To ensure fairness and prevent discrimination, it is sometimes necessary to consider counterfactual reasoning (Plecko & Bareinboim, 2022; Pearl & Mackenzie, 2018, Chapter 9). For example, in *Carson v. Bethlehem Steel Corporation* (1996) [8], the ruling stated that the core of discrimination issues lies in determining whether an individual or a group would have been treated differently by altering only the sensitive attribute (e.g., sex, age, and race) while keeping other factors constant. This is apparently a counterfactual statement. Zhang & Bareinboim (2018) introduced the SCM framework in fair decision-making problems, which provides researchers with a formal language to discuss fairness issues, particularly regarding counterfactual queries. Based on the causal language, the authors propose counterfactual direct effects, indirect effects, and spurious effects, which correspond to different types of discrimination. Further, the authors derived the causal explanation formula, which quantitatively analyzes and explains the observed disparities of decisions and helps us to examine the influence of discrimination within the decision-making process. Huang et al. (2022c) studied fairness in recommendation scenarios, focusing on the contextual bandits problem. They employed causal graphs to formally analyze the fairness concerns related to this problem, introducing a concept known as counterfactual individual fairness. Specifically, this concept involves evaluating the expected reward an individual would have received if placed in a different sensitive group. They designed an algorithm that estimates the fairness discrepancy using do-calculus. The experimental results demonstrate that their proposed approach not only enhances fairness but also maintains good recommendation performance.

Various types of fairness measures have been studied extensively in fairness literature. However, as highlighted by Kusner et al. (2017) and Zhang & Bareinboim (2018), fairness measures built upon correlations (Zafar et al., 2015; Wen et al., 2021), e.g., demographic parity and equal opportunity, do not explicitly differentiate the mechanisms through which sensitive attributes or other factors influence outcomes, which may increase discrimination in certain scenarios. Plecko & Bareinboim (2022) proposed a framework for

---

[8] https://caselaw.findlaw.com/us-7th-circuit/1304532.html

fairness analysis through a causal lens, relating the observed changes to the unobservable causal mechanisms. They articulated a principled framework to quantify and decompose fairness measures, as well as the interplay between these measures when examining fairness from a population to an individual level. The authors also provided insightful illustrations of these concepts using a range of examples. Finally, we note that there has been a growing body of research papers investigating the intersection of causality and fair decision-making in recent years (Nabi et al., 2019; Creager et al., 2020; Zhang et al., 2020c; Tang et al., 2023; Plecko & Bareinboim, 2023), but how to apply causal reinforcement learning techniques to solve such problems remains to be further explored.

## 6.3 Safety

### 6.3.1 Safety in Reinforcement Learning

Safety is a crucial concern in RL (García & Fernández, 2015; Gu et al., 2022). RL agents may sometimes act unexpectedly, especially when faced with unseen situations. This issue poses a significant risk in safety-critical applications, such as healthcare or autonomous vehicles, where even a single error could have severe consequences. Additionally, RL agents may prioritize higher returns over their own safety, known as the agent safety problem (Fulton & Platzer, 2018; Beard & Baheri, 2022). For instance, in domains like robotic control, agents may sacrifice their lifespan for a higher mission completion rate. Addressing these safety concerns and developing robust methods to ensure safe decision-making in RL are vital for the practical deployment of RL systems in real-world applications.

### 6.3.2 Safe Reinforcement Learning with Causality

Safe RL problems are typically formulated as constrained MDPs (Altman, 1995; 1999), which extend MDPs by incorporating an additional constraint set to express various safety concerns. Existing methods primarily focus on preventing constraint violations (Achiam et al., 2017; Chow et al., 2017), and seldom explicitly considering causality. Causal inference provides some valuable tools for studying safety. As an example, Hart & Knoll (2020) investigated the safety issue relates to autonomous driving. Researchers can conduct counterfactual policy evaluations before deploying any policy to the real world by utilizing counterfactual reasoning. The experimental results show that their method demonstrated a high success rate while significantly reducing the collision rate. On the other hand, Everitt et al. (2021) studied a critical concern known as reward tampering, which refers to the potential for RL agents to manipulate their reward signals. This manipulation can lead to unintended consequences and undermine the effectiveness of the learning process, thus posing a potential safety threat. In this paper, the authors presented a set of design principles aimed at developing RL agents that are robust against reward tampering, ensuring their behavior remains aligned with the intended objectives. To establish these design principles, the authors developed a causal framework supported by causal graphs, which provide a precise and intuitive understanding of the reward tampering issue. In summary, incorporating causality helps identify potential safety threats, develop preventive solutions, and trace the causes behind unexpected outcomes, thereby preventing RL agents from repeatedly breaching safety constraints. As a result, RL methods and systems can be employed safely and responsibly, reducing the risk of catastrophic consequences.

We summarize the core ideas discussed spanning sections 3 to 6 by presenting Figure 10. This schematic diagram captures the various approaches to integrating causality into the reinforcement learning process, highlighting the key components and their interactions. As we conclude this section, we recognize that while significant progress has been made in the field of causal RL, there remain important yet underexplored avenues for future research and development. Many open problems persist within the four core challenges we have discussed. In the final section of this paper, we turn our attention to the limitations and future directions of causal RL. By examining these aspects, we aim to inspire future studies and contribute to the continued advancement of causal RL, paving the way for new breakthroughs and applications.

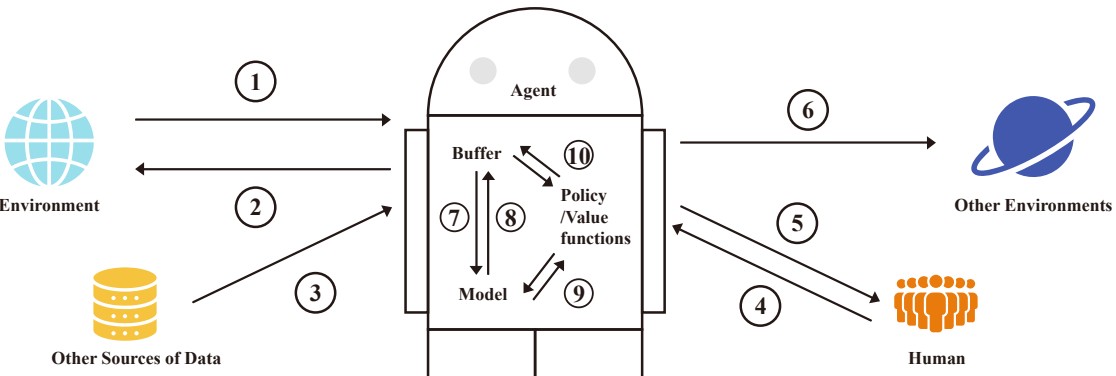

Figure 10: A schematic diagram illustrating the integration of causality into the reinforcement learning process. The numbered edges represent some key components: 1) Abstraction and extraction of causal representations from raw observations; 2) Directed exploration guided by causal knowledge; 3) Fusing (possibly confounded) data; 4) Incorporating causal assumptions or knowledge from humans. 5) Providing causality-based explanations; 6) Generalization and knowledge transfer; 7) Learning causal world models; 8) Counterfactual data generation; 9) Planning with world models; 10) Enhanced training of policies and value functions with causal reasoning.

# 7 Limitations and Future Directions

## 7.1 Limitations

So far, we have demonstrated that causal RL methods hold great promise in enhancing the decision-making capabilities of RL agents by enabling them to understand and leverage causal relationships. However, it is crucial to acknowledge the limitations associated with these methods. In this part, we will outline the limitations that researchers and practitioners may encounter when employing causal RL techniques.

One of the foremost limitations lies in the requirement for domain knowledge. Many causal RL methods heavily rely on causal graphs, thus, making accurate causal assumptions is of significance. For example, when dealing with unobserved confounding, the use of proxy variables and instrumental variables may introduce additional risks (Martens et al., 2006; Sainani, 2018; Cui et al., 2023). An inaccurate representation of the variables of interest through proxies can introduce new confounding or other forms of biases. In terms of instrumental variables, meeting all the rigorous conditions in its definition or establishing and validating these conditions in practical scenarios can be highly challenging. These limitations highlight the importance of carefully considering the underlying assumptions and the potential risks associated with the lack of knowledge.

In some real-world scenarios, the raw data is often unstructured, such as images and textual data. The causal variables involved in the data generation process may remain unknown, necessitating the development of approaches to extract causal representations from high-dimensional raw data. However, the presence of unobserved confounders may complicate the representation learning process. Learning a causal model in a latent space is generally infeasible without sufficient domain knowledge. Effectively extracting causal representations thus relies on making certain assumptions (Schölkopf et al., 2021). Learning a causally correct model with limited prior knowledge poses a significant challenge (Rezende et al., 2020). In certain scenarios, acquiring a causal model can be more demanding than directly learning the optimal policies, which may offset the sample efficiency gains.

Moreover, scaling causal RL to complex environments presents a significant challenge due to increased computational costs and model complexity. For example, the algorithm proposed by Sontakke et al. (2021) exhibits exponential growth in complexity. To simplify the learning problem, Buesing et al. (2019) assumed

Table 5: The three components of causal learning.

|  | Available information | Targets to identify | Typical questions |
|---|---|---|---|
| **Causal representation learning** | Observations | Causal variables (representation) | What factors account for the change in position? |
| **Causal discovery** | Causal variables | Causal graph | Does mass determine the change in position of an object? |
| **Causal mechanisms learning** | Causal graph | Causal mechanisms | How mass determine the change in position of an object? |

access to the true transition function and reward function in their experiments, which significantly limits the practical applicability of the proposed method.

When focusing on generalizability, we observe additional limitations. Many causal RL methods aimed at generalizability adopt causal representation learning or dynamics learning. These approaches generally require learning from multi-domain data or allowing for explicit interventions in environmental components to simulate the generation of interventional data. The quality and granularity of the learned representations closely rely on which distributional shifts, interventions or relevant signals are available (Schölkopf et al., 2021), while agents often have access to only a limited number of domains in reality.

Lastly, it is important to acknowledge the limitations associated with counterfactual reasoning. Obtaining accurate and reliable counterfactual estimates often requires making strong assumptions about the underlying causal structure (Shpitser & Pearl, 2007; Correa et al., 2021; Zhang et al., 2022b), as counterfactuals, by definition, cannot be directly observed. Some counterfactual quantities are almost never identifiable while some are identifiable under certain assumptions, such as the effect of treatment on the treated (ETT) (Pearl, 2009b, Chapter 11). Furthermore, the computational complexity of counterfactual reasoning can be a bottleneck, especially when dealing with high-dimensional state and action spaces. This complexity can hinder real-time decision-making in complex tasks, which remains an ongoing challenge.

### 7.2 Causal Learning in Reinforcement Learning

In section 3.3 and section 5.2, we explained how causal dynamics learning - a class of methods closely related to MBRL - can improve generalizability and sample efficiency (Wang et al., 2022; Huang et al., 2022b). These methods focus on understanding the cause-and-effect relationships between variables and the process that generates these variables. Instead of using complex, redundant connections, to model the data generation process, these methods prefer a sparse, modular style. As a result, they are more efficient and stable than traditional model-based methods and allow RL agents to adapt quickly to unseen environments or tasks. However, we may not have perfect knowledge of the causal variables in reality. Sometimes, we must deal with high-dimensional and unstructured data like visual information. In this case, RL agents need to be able to extract causal representations from raw data (Schölkopf et al., 2021). Depending on the tasks of interest, causal representations can take various forms, ranging from abstract concepts like emotions and preferences to more concrete entities such as physical objects.

The complete process of learning a causal model from raw data is known as causal learning (Peters et al., 2017). It is different from causal reasoning (Imbens & Rubin, 2015; Glymour et al., 2016), which only focuses on estimating specific causal effects given the causal model. Causal learning involves extracting causal representation, discovering causal relationships, and learning causal mechanisms. Table 5 briefly summarizes their characteristics. All three of these components are significant and deserve further investigation. A great deal of research has been done on causal discovery (Spirtes et al., 2000; Pearl, 2009b; Peters et al., 2017; Vowels et al., 2022), a process of recovering the causal structure of a set of variables from data, particularly concerning conditional independence tests (Spirtes et al., 2000; Sun et al., 2007; Hoyer et al., 2008; Zhang et al., 2011). Under certain assumptions, such as faithfulness, algorithms can identify the Markov equivalence class of the underlying causal graph from observational data. It is very difficult to uniquely identify a causal graph from observational data without strong assumptions about the data generation process. Therefore, some research efforts in causal inference have been done on partially identifiable causal graphs, such as maximally oriented partially directed acyclic graphs (MPDAGs) (Meek, 1995; Perkovic et al., 2017; Perkovic,

2020). With additional background knowledge, we can identify more causal directions, going beyond Markov equivalence classes. Nonetheless, there remains a dearth of research on effectively leveraging these partially directed graphs to address research challenges in reinforcement learning. This represents a promising area for future exploration. In addition, combining RL with causal discovery empowers an agent to actively gather interventional data from the environment to recover the underlying causal structure. This opens up an intriguing research direction that focuses on exploring methods for leveraging interventional data, or a combination of observational and interventional data, to facilitate efficient causal discovery (Addanki et al., 2020; Jaber et al., 2020; Brouillard et al., 2020; Zhu et al., 2022a).

As for causal representation learning (Schölkopf et al., 2021; Wang & Jordan, 2022; Shen et al., 2022), one possible solution is to learn latent factors from high-dimensional observations using autoencoders (Yang et al., 2021a; Eghbal-zadeh et al., 2021; Tran et al., 2022). These methods can approximatively recover causal representations and structures by virtue of carefully designed constraint terms. This idea inherently embeds an SCM into the learner, implicitly binding causal discovery and causal mechanisms learning in one solution. Additionally, in scenarios involving multiple environments or tasks, causal representations can also be derived through techniques like mining invariance (Zhang et al., 2020a; Bica et al., 2021b; Saengkyongam et al., 2022) or clustering trajectories from diverse domains (Sontakke et al., 2021). However, determining the optimal number and granularity of causal variables remains challenging, as the optimal causal representations often depend on the specific task.

Overall, causal learning in RL is an underexplored problem and has the potential to advance the RL community. Additionally, RL techniques show promise in contributing to the field of causal learning. As we delve deeper into the connections between these two fields, an exchange of insights can nurture reciprocal benefits, propelling both fields forward (Zhu et al., 2022a).

### 7.3   Causality-aware Multitask and Meta Reinforcement Learning

Multitask reinforcement learning (Parisotto et al., 2015; Teh et al., 2017; D'Eramo et al., 2020; Vithayathil Varghese & Mahmoud, 2020) focuses on simultaneously learning multiple tasks by sharing knowledge and leveraging synergies across tasks. This scenario commonly arises in robot manipulation, where a robot needs to acquire various skills, such as reaching, grasping, and pushing. Meta-learning (Duan et al., 2016; Finn et al., 2017; Gupta et al., 2018; Xu et al., 2018), on the other hand, involves training on a task distribution to gain the ability to adapt quickly to a new task. Impressive results have been achieved without explicitly considering causality, leading to the question: Is training a high-capacity model on a diverse range of tasks sufficient to generalize to new tasks? Recent research empirically validates this hypothesis, as large pre-trained models on diverse datasets have demonstrated remarkable performance in tasks requiring few-shot learning or even zero-shot generalization ability (Brown et al., 2020; Wei et al., 2021).

As shown in Figure 5, different tasks are essentially different interventions in the data generation process (Schölkopf et al., 2021), so it is not surprising that models trained on multiple tasks can achieve excellent generalizability – they implicitly learn the causal relationships governing the data generation process of these tasks. Dasgupta et al. (2018) provide compelling evidence for this proposition. Their study demonstrates that the capability of causal inference may emerge from large-scale meta-RL. Nonetheless, testing all possible interventions and their combinations in real-world scenarios is impractical. This is where causal modeling comes into play. As discussed in the previous section, causal models enable the explicit incorporation of prior knowledge, helping agents to align their understanding of the world with human cognition. Moreover, causal relationships facilitate problem abstraction (Wang et al., 2022), thereby eliminating the need of testing irrelevant interventions. Additionally, agents that organize their knowledge using causal structures benefit from the stability of causal relationships. While non-causal agents would require finetuning the whole model for even a slightly changed task, a causality-aware agent would only need to adjust a few modules (Huang et al., 2022a), exhibiting a stronger knowledge transfer ability. These properties may also contribute to lifelong (or continual) learning (Xie & Finn, 2022; Khetarpal et al., 2022), allowing for fast adaptation to new tasks that arise in sequence.

### 7.4 Human-in-the-loop Learning and Reinforcement Learning from Human Feedback

Human-in-the-loop learning (HiLL) (Mosqueira-Rey et al., 2022) is a form of machine learning in which humans actively participate in the development cycle of machine learning models or algorithms. This can involve providing labels, preferences, or other types of feedback. When the data or task being learned is complex or requires high levels of cognition, HiLL often produces better results because humans can provide valuable insights or knowledge to the model that it may be difficult for the model to learn on its own (Mosqueira-Rey et al., 2022; Zhang & Bareinboim, 2022a).

In the context of RL, HiLL typically involves incorporating human expertise into the MDP to replace the reward function that provides feedback signals. This approach allows us to train RL agents with the help of human knowledge and values, alleviating the challenges associated with defining a sophisticated reward function (Zhang & Bareinboim, 2022a). This idea is closely related to RLHF (Reinforcement Learning from Human Feedback) (Christiano et al., 2017), a concept that has gained increasing attention recently in the training of large language models (Ziegler et al., 2020; Glaese et al., 2022; Ouyang et al., 2022), where human instructors provide rewards (or penalties) to a model to encourage (or discourage) certain behaviors. From a causal perspective, humans can provide machine learning models with a strong understanding of causality based on their knowledge of the world, which can help filter out behaviors that may lead to negative outcomes. However, it is important to note that humans and machines may have different observations or perceptions of the world, and non-causal-aware RL agents may be influenced by confounding variables (Gasse et al., 2023). In addition, we often need to consider the issue of limited budgets, as our goal is to provide meaningful feedback to RL agents at the lowest possible cost. Finally, in addition to scalar feedback, we may also provide more informative feedback to agents in the form of counterfactuals (Karalus & Lindner, 2022).

### 7.5 Theoretical Advances in Causal Reinforcement Learning

Theoretical research in causal RL has mainly concentrated on the MAB problem. Lattimore et al. (2016) first introduced the causal bandit problem, where interventions are treated as arms, and their impact on the reward and other observable covariates is associated with a known causal graph. Unlike contextual bandits, in this setting, the observations occur after each intervention is made. Using the structural knowledge from the causal model, the agent can efficiently identify good interventions, achieving strictly better regret than algorithms that lack causal information. Sen et al. (2017) further considered incorporating prior knowledge about the strength of interventions. Since different arms (interventions) share the same causal graph, samples from one arm can inform us about the expected value of other arms. The authors demonstrated significant theoretical and practical improvements by leveraging such information leakage.

However, these works only focused on intervening in a single node on the causal graph, where causal effects propagate only to the first-order neighbors of the intervened node. Yabe et al. (2018) generalize this setting by studying an arbitrary set of interventions, allowing for causal effects to propagate throughout the causal graph. Lee & Bareinboim (2018) also investigated the scenario of pulling arms in a combinatorial manner. They showed that considering all interventable variables as one arm may lead to a suboptimal policy, and the structural information can be used to identify the minimal intervention set that is worth intervening in. Lu et al. (2021) improved the naïve bounds derived in Lattimore et al. (2016), devising algorithms for causal bandit problems with sublinear cumulative regret bounds. Nair et al. (2021) study the causal bandit problem with budget constraints. In this setting, there is a trade-off between the more economical observational arm (i.e., no interventions on the causal graph) and the high-cost interventional arms. While the observational arm is less rewarding, it offers valuable information for studying other arms at a significantly lower cost. More recently, Kroon et al. (2022) introduced the concept of "separating set" from causal discovery to causal bandit problems, which renders a target variable independent of a context variable when conditioned upon. By utilizing this separating set, the authors developed a bandit algorithm that does not rely on the assumption of prior causal knowledge. On the other hand, Bilodeau et al. (2022) studied the adaptivity issue concerning the d-separator, i.e., whether an algorithm can perform nearly as well as an algorithm with oracle knowledge of the presence or absence of a d-separator.

While causal bandit problems have yielded fruitful results, there have been notably fewer studies dedicated to MDP problems. As discussed in Section 4.3, some studies focused on DTRs (Zhang & Bareinboim,

2019; 2020), which can be modeled as an MDP with a global confounder variable. Additionally, there have been some investigations into confounded MDP (Zhang & Bareinboim, 2016; Wang et al., 2021b), which extend the concept of DTR by admitting the presence of unobserved confounders at each time step. In the theory of causal RL, aside from understanding how causal information, especially causal graphs, enhance the regret bounds, the identifiability of causal effects (Zhang et al., 2020b; Lu et al., 2022) and structure (Huang et al., 2022a) are also of great interest. While existing research provides valuable insights into the theoretical foundations of causal RL, further theoretical advancements are necessary. In addition to helping us comprehend the reasons behind the success of existing causal RL methods, we also hope that future theoretical advances will pave the way for designing novel and effective approaches that are both interpretable and robust for real-world applications.

### 7.6  Benchmarking Causal Reinforcement Learning

In RL, we are typically interested in the efficiency and convergence of the algorithm. Atari 2600 Games and Mujoco locomotion tasks (Brockman et al., 2016) are commonly used as benchmarks for discrete and continuous control problems. There are also experimental environments that evaluate the generalizability and robustness of RL, such as Procgen (Cobbe et al., 2020). Some benchmarks focus on multitask learning, meta-learning, and curriculum learning for reinforcement learning, such as RLBench (James et al., 2019), Meta-World (Yu et al., 2021), Alchemy (Wang et al., 2021a), and Causal-World Ahmed et al. (2022). Among them, Causal-World offers a wide range of robotic manipulation tasks that share a common attribute set and structure. In these tasks, the robot is required to construct a goal shape using the provided building blocks. One notable feature of this benchmark is its provision of interfaces that allow manual modification of object attributes such as size, mass, and color. This enables researchers to intervene in these attributes and generate a series of tasks with consistent causal structures.

Since causal RL is not limited to a particular type of problem, evaluation metrics may vary depending on the specific mission. While existing experimental environments have provided good benchmarks for evaluating algorithms with various metrics, the underlying data generation processes in these environments often remain opaque, concealed within game simulators or physics engines. This lack of transparency makes it difficult for researchers to fully understand the causal mechanisms behind the problems they are attempting to solve, hindering the development of the field. Recently, Ke et al. (2021) proposed a new set of environments focusing on causal discovery in visual-based RL, allowing researchers to specify the causal graph and customize its complexity. Nevertheless, as we demonstrated in sections 5 to 6, a large portion of the existing research still relies on toy environments to evaluate the effectiveness of algorithms. Furthermore, in addition to the previously mentioned properties, a good benchmark should consider the multiple factors comprehensively, as discussed in section 6. Thus, developing a comprehensive benchmark for assessing the performance of causal RL methods remains an ongoing challenge.

### 7.7  Real-world Causal Reinforcement Learning

Finally, it is worth noting that the practical implementation of causal RL in real-world applications remains limited. To make it more widely applicable, we must carefully examine the challenges posed by reality. Dulac-Arnold et al. (2020; 2021) identified nine critical challenges that are holding back the use of RL in the real world: limited samples; unknown and large delays; high-dimensional input; safety constraints; partial observability; multi-objective or unspecified reward functions; low latencies; offline learning; and the need for explainability.

We have discussed some of these issues throughout this paper, many of which are related to ignorance of causality. For instance, the challenge of learning from limited samples corresponds to the sample efficiency issue discussed in section 5.2. Learning from high-dimensional inputs and multiple reward functions relates to the generalization problem outlined in section 3.3. Offline learning raises concerns about spurious correlations (section 4.3), and security and explainability are covered in section 6.

Although causal models offer promising solutions to these real-world challenges, current experimental environments often fall short of meeting the research needs. As discussed in section 7.6, popular benchmarks are often treated as black boxes, and researchers have limited access to and understanding of the causal

mechanisms by which these black boxes generate data. This lack of transparency significantly hinders the development of this research field. In order to facilitate the widespread adoption and application of causal RL, it is crucial to address these limitations and cultivate a culture of causal thinking.

## 8 Conclusion

In summary, Causal RL holds promise for tackling complex decision-making problems under uncertainty. It represents an understudied yet significant research direction. By explicitly integrating assumptions or knowledge about the causal relationship underlying the data generation process, causal RL algorithms can learn optimal policies more efficiently and make more informed decisions. In this survey, we aimed to clarify the terminologies and concepts related to causal RL and to establish connections between existing work. We proposed a problem-oriented taxonomy and systematically discussed and analyzed the latest advances in causal RL, focusing on how they address the four critical challenges facing RL.

While there is still much work to be done in this field, the results to date are encouraging. They suggest that causal RL has the potential to significantly improve the performance of RL systems in a wide range of problems. Here, we summarize the key conclusions of this survey.

- Causal RL is an emerging branch of RL that emphasizes understanding and utilizing causality to make better decisions (section 2.3).

- Causal modeling has the potential to enhance generalization (section 3.3) and sample efficiency (section 5.2), yet it comes with fundamental challenges and limitations that require careful consideration (section 7.1). With limited causal information, RL agents may need to learn about causal representation and environmental dynamics from raw data (section 7.2).

- Causal effects and relationships are identifiable under certain conditions (section 2.3 and section 7.5), enabling agents to effectively utilize observational data. Furthermore, multitask learning and meta-learning may help facilitate causal learning (section 7.3) In turn, harnessing causality can enhance the ability to transfer knowledge and effectively address a wide range of tasks (section 3.3).

- Correlation does not imply causation. Spurious correlations can lead to a distorted understanding of the environment and task, resulting in suboptimal policies (section 4.3). In addition to employing causal reasoning techniques, leveraging human understanding of causality may further enhance reinforcement learning (section 7.4).

- In real-world applications, performance is not the only concern. Factors such as explainability, fairness, and security, must also be considered (section 6). Current benchmarks call for increased transparency and a comprehensive, multi-faceted evaluation protocol for reinforcement learning (section 7.6), which has significant implications for advancing real-world applications of causal reinforcement learning (section 7.7).

We hope this survey will help establish connections between existing work in causal reinforcement learning, inspire further exploration and development, and provide a common ground and comprehensive resource for those looking to learn more about this exciting field.

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

# A Supplementary Introduction to Causality

## A.1 Mediation analysis

Mediation is a causal concept that closely relates to counterfactuals. The goal of mediation analysis is to examine the direct or indirect effects of a treatment variable [9] $X$ on an outcome variable $Y$, mediated by a third variable $M$ (referred to as the mediator). To illustrate, let's consider the direct effect of different food consumption on preventing scurvy (See Figure 3). In a counterfactual world, if we prevent changes in the mediator variable $M$ (e.g., vitamin C intake), then whatever changes the outcome variable $Y$ (e.g., whether a sailor gets scurvy) can only be attributed to changes in the treatment variable $X$ (e.g., the food consumed), allowing us to establish the observed effect as a direct effect of $X$ on $Y$. It is worth noting that, in cases like this, the statistical language can only provide the conditioning operator, which merely shifts our attention to individuals with equal values of $M$. On the other hand, the do-operator precisely captures the concept of keeping the mediator $M$ unchanged. These two operations lead to fundamentally different results, and conflating them can yield opposite conclusions (Pearl & Mackenzie, 2018, Chapter 9). In summary, such analyses are crucial for understanding the potential causal mechanisms and paths in complex systems, with applications spanning various fields including psychology, sociology, and epidemiology (Carey & Wu, 2022).

## A.2 Causal discovery and causal reasoning

. In the field of causal inference, there are two primary areas of focus: causal discovery and causal reasoning. Causal discovery involves inferring the causal relationships between variables of interest (in other words, identifying the causal graph of the data generation process). Traditional approaches use conditional independence tests to infer causal relationships, and recently some studies have been conducted based on large datasets using deep learning techniques. Glymour et al. (2019) and Vowels et al. (2022) comprehensively survey the field of causal discovery.

As opposed to causal discovery, causal reasoning investigates how to estimate causal effects, such as intervention probability, given the causal model. Interventions involve actively manipulating the system or environment, which can be costly and potentially dangerous (e.g., testing a new drug in medical experiments). Therefore, a core challenge of causal reasoning is how to translate causal effects into estimands that can be estimated from observational data using statistical methods. Given the causal graph, the identifiability of causal effects can be determined systematically through the use of do-calculus (Pearl, 1995).

## A.3 Causal representation learning

A fundamental limitation of traditional causal inference is that most research starts with the assumption that causal variables are given, which does not align with the reality of dealing with high-dimensional and low-level data, such as images, in our daily lives. Causal representation learning (Schölkopf et al., 2021) is an emerging field dedicated to addressing this challenge. Specifically, causal representation learning focuses on identifying high-level variables from low-level observations. These high-level variables are not only descriptive of the observed data but also explanatory of the data generation process, as they capture the underlying causal mechanisms. By effectively discovering meaningful and interpretable high-level variables, causal representation learning facilitates causal inference in complex, high-dimensional domains.

# B A Brief Introduction to Environments and Tasks

In this section, we briefly introduce the environments and tasks mentioned spanning sections 3 to 6.

---

[9] A treatment variable, also known as an intervention variable, refers to a variable that is purposefully manipulated to assess its impact on one or more outcome variables of interest.

### B.1 Autonomous Driving

**BARK-ML**: `https://github.com/bark-simulator/bark-ml`. BARK is an open-source behavior benchmarking environment. It covers a full variety of real-world, interactive human behaviors for traffic participants, including Highway, Merging, and Intersection (Unprotected Left Turn).

**Crash (highway-env)**: `https://github.com/eleurent/highway-env`. The highway-env project gathers a collection of environments for autonomous driving and tactical decision-making tasks, including Highway, Merge, Roundabout, Parking, Intersection, and Racetrack. Crash is a modified version by (Ding et al., 2022) that is not publicly available.

### B.2 Classical Control

**Cart-pole (dm_control)**: `https://github.com/svikramank/dm_control`. Cart-pole is an environment belonging to the DeepMind Control Suite. It involves swinging up and balancing an unactuated pole by applying forces to a cart at its base.

**Acrobot (OpenAI Gym)**: `https://www.gymlibrary.dev/environments/classic_control/acrobot/`. The Acrobot environment consists of two links connected linearly to form a chain, with one end of the chain fixed. The goal is to apply torques on the actuated joint to swing the free end of the linear chain above a given height while starting from the initial state of hanging downwards.

**Mountain Car (OpenAI Gym)**: `https://www.gymlibrary.dev/environments/classic_control/mountain_car/`. The Mountain Car MDP is a deterministic MDP that consists of a car placed stochastically at the bottom of a sinusoidal valley, with the only possible actions being the accelerations that can be applied to the car in either direction. The goal of the MDP is to strategically accelerate the car to reach the goal state on top of the right hill.

**Cart Pole (OpenAI Gym)**: `https://www.gymlibrary.dev/environments/classic_control/cart_pole/`. A pole is attached by an un-actuated joint to a cart, which moves along a frictionless track. The pendulum is placed upright on the cart and the goal is to balance the pole by applying forces in the left and right direction on the cart.

**Pendulum (OpenAI Gym)**: `https://www.gymlibrary.dev/environments/classic_control/pendulum/`. The system consists of a pendulum attached at one end to a fixed point, and the other end being free. The pendulum starts in a random position and the goal is to apply torque on the free end to swing it into an upright position, with its center of gravity right above the fixed point.

**Inverted Pendulum (OpenAI Gym)**: `https://www.gymlibrary.dev/environments/mujoco/inverted_pendulum/`. This environment involves a cart that can moved linearly, with a pole fixed on it at one end and having another end free. The cart can be pushed left or right, and the goal is to balance the pole on the top of the cart by applying forces on the cart.

### B.3 Game

**MiniPacman**: `https://github.com/higgsfield/Imagination-Augmented-Agents`. MiniPacman is played in a $15 \times 19$ grid-world. Characters, the ghosts and Pacman, move through a maze.

**Lunar Lander (OpenAI Gym)**: `https://www.gymlibrary.dev/environments/box2d/lunar_lander/`. This environment is a classic rocket trajectory optimization problem. The landing pad is always at coordinates $(0, 0)$. The coordinates are the first two numbers in the state vector. Landing outside of the landing pad is possible. Fuel is infinite, so an agent can learn to fly and then land on its first attempt.

**Bipedal Walker (OpenAI Gym)**: `https://www.gymlibrary.dev/environments/box2d/bipedal_walker/`. This is a simple 4-joint walker robot environment.

**Car Racing (OpenAI Gym)**: `https://www.gymlibrary.dev/environments/box2d/car_racing/`. Car Racing is a top-down racing environment. The generated track is random in every episode. Some indicators

are shown at the bottom of the window along with the state RGB buffer. From left to right: true speed, four ABS sensors, steering wheel position, and gyroscope.

**Beam Rider (OpenAI Gym)**: `https://www.gymlibrary.dev/environments/atari/beam_rider/`. The agent controls a space-ship that travels forward at a constant speed. The agent can only steer it sideways between discrete positions. The goal is to destroy enemy ships, avoid their attacks and dodge space debris.

**Pong (OpenAI Gym)**: `https://www.gymlibrary.dev/environments/atari/pong/`. The agent controls the right paddle, competing against the left paddle controlled by the computer.

**Pong (Roboschool)**: `https://github.com/openai/roboschool`. Roboschool is an open-source software for robot simulation, which is now deprecated. Pong allows for multiplayer training.

**Sokoban**: `https://github.com/mpSchrader/gym-sokoban`. This game is a transportation puzzle, where the player has to push all boxes in the room on the storage locations/ targets. The possibility of making irreversible mistakes makes these puzzles so challenging especially for RL algorithms.

**SC2LE**: `https://github.com/deepmind/pysc2`. SC2LE is a RL environment based on the StarCraft II game. It is a multi-agent problem with multiple players interacting. This domain poses a grand challenge raising from the imperfect information, large action and state space, and delayed credit assignment.

**VizDoom**: `https://github.com/Farama-Foundation/ViZDoom`. ViZDoom is based on Doom, a 1993 game. It allows developing AI bots that play Doom using only visual information.

## B.4 Healthcare

**MIMIC-III**: `https://physionet.org/content/mimiciii/1.4/`. MIMIC-III (Johnson et al., 2016) is a large, freely-available database comprising deidentified health-related data associated with over forty thousand patients who stayed in critical care units of the Beth Israel Deaconess Medical Center between 2001 and 2012. Lu et al. (2020) use the code (`https://github.com/aniruddhraghu/sepsisrl`) provided by Raghu et al. (2017) for preprocessing the data and Bica et al. (2021b) open-source their code on `https://github.com/vanderschaarlab/Invariant-Causal-Imitation-Learning/tree/main`.

**Therapy**: `https://github.com/Networks-Learning/counterfactual-explanations-mdp/blob/main/data/therapy/README.md`. This dataset contains real data from cognitive behavioral therapy. The data were collected during a clinical trial with the patients' written consent A post-processed version of the data is available upon request from `Kristina.Fuhr@med.uni-tuebingen.de`.

**Sepsis**: `https://github.com/clinicalml/gumbel-max-scm`. This environment, originally developed by Oberst & Sontag (2019) to simulate intensive care trajectories, is adopted by Pace et al. (2023) to investigate the impact of hidden confounders in offline RL. The state space comprises 4-dimensional normalized observation vectors (heart rate, systolic blood pressure, oxygenation, and blood glucose levels), and the discrete action space encompasses three binary treatments (antibiotic, vasopressor, and ventilation). In addition, the patient's diabetic status serves as an unobserved binary variable in this environment.

**HiRID**: `https://physionet.org/content/hirid/1.1.1/`. The HiRID dataset (Hyland et al., 2020; Faltys et al., 2021), comprising data from over 34,000 patient admissions at the Bern University Hospital's Department of Intensive Care Medicine in Switzerland, offers a high-resolution collection of demographic, physiological, diagnostic, and treatment information. Pace et al. (2023) focused on optimizing fluid and vasopressor administration to prevent circulatory failure in their research targeted at non-identifiable hidden confounding in offline RL.

## B.5 Robotics

### B.5.1 Locomotion

**Cheetah (dm_control)**: `https://github.com/svikramank/dm_control`. Cheetah is an environment belonging to the DeepMind Control Suite. It is a running planar bipedal robot.

**OpenAI Gym**: `https://www.gymlibrary.dev/environments/mujoco/`. These environments are built upon the MuJoCo (Multi-Joint dynamics with Contact) engine. The goal is to make the 3D robots move in the forward direction by applying torques on the hinges connecting the links of each leg and the torso.

**PyBullet Gym**: `https://github.com/benelot/pybullet-gym`. This is an open-source implementation of OpenAI Gym MuJoCo environments using the Bullet Physics (`https://github.com/bulletphysics/bullet3`).

**D4RL**: `https://github.com/Farama-Foundation/D4RL`. D4RL is an open-source benchmark for offline RL. It includes several OpenAI Gym benchmark tasks, such as the Hopper, HalfCheetah, and Walker environments.

### B.5.2 Manipulation

**CausalWorld**: `https://github.com/rr-learning/CausalWorld`. CausalWorld is an open-source simulation framework and benchmark for causal structure and transfer learning in a robotic manipulation environment where tasks range from rather simple to extremely hard. Tasks consist of constructing 3D shapes from a given set of blocks - inspired by how children learn to build complex structures.

**Isaac Gym**: `https://github.com/NVIDIA-Omniverse/IsaacGymEnvs`. Isaac Gym offers a high performance learning platform to train policies for wide variety of robotics tasks directly on GPU.

**OpenAI**: The origin version is developed by OpenAI, known as "Ingredients for robotics research" (`https://openai.com/research/ingredients-for-robotics-research`), and now is maintained by the Farama Foundation (`https://github.com/Farama-Foundation/Gymnasium-Robotics`). It contains eight simulated robotics environments.

**robosuite**: `https://github.com/ARISE-Initiative/robosuite`. robosuite is a simulation framework powered by the MuJoCo physics engine for robot learning. It contains seven robot models, eight gripper models, six controller modes, and nine standardized tasks. It also offers a modular design for building new environments with procedural generation.

### B.6 Navigation

**Unlock (Minigrid)**: `https://github.com/Farama-Foundation/MiniGrid`. The Minigrid library contains a collection of discrete grid-world environments to conduct research on Reinforcement Learning. Unlock is task designed by (Ding et al., 2022), which is not publicly available.

**Contextual-Gridworld**: `https://github.com/eghbalz/contextual-gridworld`. Agents are trained on a group of training contexts and are subsequently tested on two distinct sets of testing contexts within this environment. The objective is to assess the extent to which agents have grasped the causal variables from the training phase and can accurately deduce and extend to new (test) contexts.

**3D Maze (Unity)**: `https://github.com/Harsha-Musunuri/Shaping-Agent-Imagination`. This environment is built on the Unity3d game development engine. It contains an agent that can move around. The environment automatically changes to a new view after every episode.

**Spriteworld**: `https://github.com/deepmind/spriteworld`. Spriteworld is an environment that consists of a 2D arena with simple shapes that can be moved freely. The motivation was to provide as much flexibility for procedurally generating multi-object scenes while retaining as simple an interface as possible.

**Taxi**: `https://www.gymlibrary.dev/environments/toy_text/taxi/`. There are four designated locations in the grid world. When the episode starts, the taxi starts off at a random square and the passenger is at a random location. The taxi drives to the passenger's location, picks up the passenger, drives to the passenger's destination (another one of the four specified locations), and then drops off the passenger. Once the passenger is dropped off, the episode ends.

### B.7 Others

**Chemical**: `https://github.com/dido1998/CausalMBRL#chemistry-environment`. By allowing arbitrary causal graphs, this environment facilitates studying complex causal structures of the world. This is illustrated through simple chemical reactions, where changes in one element's state can cause changes in the state of another variable.

**Light**: `https://github.com/StanfordVL/causal_induction`. It consists of the light switch environment for studying visual causal induction, where $N$ switches control $N$ lights, under various causal structures. Includes common cause, common effect, and causal chain relationships.

**MNIST**: `http://yann.lecun.com/exdb/mnist/`. The MNIST dataset contains 70,000 images of handwritten digits.

