# OpenReview forum: "Causal Reinforcement Learning: A Survey"
_TMLR — Accepted by TMLR_

### Review · Reviewer_nERm · 2023-08-30

**Summary Of Contributions:**

This survey paper investigates the intersection of two important and promising areas, reinforcement learning and causality. The motivation of this combination is that reinforcement learning agents lack a fundamental understanding of the world and must therefore learn from scratch through numerous trial-and-error interactions, which could probably be fixed by causality that formalizes knowledge in a systematic manner and leverages invariance for effective knowledge transfer. Particularly, the contribution can be summarized as below:
* explores the improvements of Causal RL over traditional RL methods.
* identifies key challenges in RL that can be effectively addressed by causality.
* provides a categorization of existing Causal RL methods.
* summarizes major unresolved issues and promising research directions in causal RL.

**Audience:**

Yes

**Broader Impact Concerns:**

There is no ethical concern as far as I know.

**Claims And Evidence:**

Yes

**Requested Changes:**

Please see points 1-4 of the weaknesses.

**Strengths And Weaknesses:**

**Strengths:**
1. There are not many similar survey papers that investigate the combination of causality and RL at the submission date of this manuscript. This work should have high empirical value to inspire future RL works.
2. The example of the cause of scurvy is intuitive. Using this example throughout the entire paper makes many concepts easy to understand.
3. The comparison and categorization of existing works is a good summarization. This will be helpful to future works that seek to find comparable baselines.

**Weaknesses:**
1. More intuitive examples about causal RL. The cause of the scurvy example still belongs to the causality field as there is no sequential decision-making problem. I think more concrete examples about when causality appears in RL tasks and how causality influences the decisions could make this paper attractive to broader readers. Maybe add one figure with several tasks and explanations.
2. Section 2.1 is too long (a “brief” introduction with 3.5 pages). As this survey targets an advanced topic of the intersection between causality and RL, it should assume the readers have basic background knowledge. The authors can refer the readers to other references, which I believe there is a lot in both RL and causality fields. Another reason I think the background of causality should be short is that Causal RL mainly focuses on how to use causality tools in RL, so it would be better to move some contents to later sections and explain causality concepts under RL contexts.
3. The specific advantages of Causal RL should be mentioned in the abstract to highlight the main contribution. I think the most valuable part of this paper is the following three sections:
    * Enhancing Sample Efficiency through Causal Reinforcement Learning
    * Advancing Generalizability and Knowledge Transfer through Causal Reinforcement Learning.
    * Addressing Spurious Correlations through Causal Reinforcement Learning

   These sections greatly summarize three advantages of using causality in RL, so maybe the authors should consider briefly mentioning them in the abstract.

4. The title of section 6 is a little confusing. The connection between this section and previous sections is also not clear. This section itself is easy to understand but the title and structure may need to be revised. For example, a similar title to previous sections: “Addressing Explainability, Fairness, and Safety through Causal RL”.
5. Lastly, I have a clarification question about the difference between spurious correlation and generalizability. As discussed in this paper [1], spurious correlation can be treated as one type of distribution shift and can be solved with the distributional robust framework. One example is that the compositional generalization can be caused by an unobserved confounder, i.e., when the value of the confounder is different, the environment gives different types of combinations of tasks. Does the author think these two advantages can be merged together?

[1] Wenhao Ding, Laixi Shi, Yuejie Chi, and Ding Zhao. "Seeing is not Believing: Robust Reinforcement Learning against Spurious Correlation." arXiv preprint arXiv:2307.07907 (2023).

---

> ### Author Response · Authors · 2023-10-01
> **Response to Reviewer nERm**
>
> Thank you very much for your thorough review and insightful feedback on our manuscript! We have carefully addressed each of your points and made necessary revisions, as outlined below:
>
> > More intuitive examples about causal RL.
>
> We have included a new figure (figure 1 on page 2) that demonstrates the intersection of causality and RL by presenting real-world scenarios (robotic manipulation, medical treatment, and autonomous driving) in which causal relationships emerge naturally from decision-making processes. We believe it can enhance the paper's accessibility to a wider range of readers, making the concepts discussed in this paper more attractive.
>
> > Section 2.1 is too long (a "brief" introduction with 3.5 pages).
>
> To provide a more concise overview in Section 2.1, we have carefully restructured this section by selectively moving some contents to later sections and discussing them under RL contexts. To be self-contained, we have also included a supplementary introduction to causality is in Appendix A.
>
> > The specific advantages of Causal RL should be mentioned in the abstract.
>
> Thank you for the thoughtful comment! We have revised the abstract accordingly.
>
> > The title of section 6 is a little confusing.
>
> We have revised the title of section 6 to "Promoting Explainability, Fairness, and Safety with Causal RL," which accurately represents the content and maintains consistency with the earlier sections.
>
> > Question about the difference between spurious correlation and generalizability.
>
> This is indeed an intriguing question worthy of further investigation! We agree that there is a subtle connection between spurious correlation and generalization, since agents may be expected to be robust against factors that raise spurious correlations when studying the generalization problem. However, these two research directions have different focuses and technique stacks, as can be intuitively seen by comparing the "Techniques" column in Tables 1 and 2. Generalization is usually performance-oriented, focusing on the performance of learned policies on new domains. On the other hand, spurious correlation often focuses on whether and how the causal effect of interest can be recognized. In recent years, a burgeoning research direction known as stable learning[1][2] has emerged to explore the intersection of these two research directions. We have cited the reference you mentioned and added a related discussion at the end of Section 4 (the last paragraph on page 18).
>
> We would like to express our gratitude once again for your valuable insights, which have undoubtedly enriched our work.
>
> [1] Zhang, Xingxuan, et al. "Deep stable learning for out-of-distribution generalization." _Proceedings of the IEEE/CVF Conference on Computer Vision and Pattern Recognition_. 2021.
>
> [2] Cui, Peng, and Susan Athey. "Stable learning establishes some common ground between causal inference and machine learning." _Nature Machine Intelligence_ 4.2 (2022): 110-115.

---

### Review · Reviewer_64gv · 2023-09-05

**Summary Of Contributions:**

Overall, I read with interest the survey paper on causal reinforcement learning. In my assessment, the paper does a pretty good job at covering various aspects of the field. The paper is well-written, informed, and it is clear that a substantial amount of effort has gone into it.

**Audience:**

Yes

**Broader Impact Concerns:**

N/A.

**Claims And Evidence:**

Yes

**Requested Changes:**

See strengths and weaknesses.

**Strengths And Weaknesses:**

I have a number of suggestions that may be considered. Some of these are mistakes that need to be fixed, and there is a number of them. To contextualize, I am somewhat familiar with the RL literature, and quite familiar with the causality literature:

  * (C1) The Book of Why (TBOW) is cited repeatedly across the paper. This is fine, but I do wish to remark that the Causality book is the canonical reference here. TBOW was intended as a less formal treatment for readers from applied sciences. I think pointers to appropriate Chapters in Causality would be more useful than citations of TBOW.

  * (C2) The scurvy example in the second paragraph is somewhat difficult to comprehend. From the current writing, my reading is that "knowing how to treat a disease requires knowing what causes it", which seems somewhat of an obvious statement. I believe this paragraph could be improved, in particular highlighting how various sentences relate to a formal causal model.

  * (C3) Relating to (C2), I also think paragraph on the distinction between RL and Causal RL could be improved (page 2, 2nd paragraph). It seems that CRL reduces the size of the action space that is interesting for exploration. However, in my view, the scurvy example currently does not convey this message very clearly.

  * (C4) Footnote in the SCM paragraph: It is really unusual to state that the Markovian assumption is considered a consensus in causal inference. This is genuinely not the case, witnessed by the huge amount of literature on Semi-Markovian models, and inference in presence of latent confounding. This footnote should be removed altogether.

  * (C5) Product decomposition paragraph: V_i is not independent of V \setminus {V_i, pa(V_i), ch(V_i)} given pa(V_i). There could easily be descendants of V_i that are not its children, and conditioning on pa(V_i) does not block any paths to the descendants. This sentence is unsual, and sounds like a description of Markov blanket, which is not relevant here. This part of the manuscript should be revised.

  * (C6) Distributions P(v_i | pa(v_i)) are usually called Markov kernels. Causal mechanisms are usually reserved for the structural functions f_i. Please double check the Causality book.

  * (C7) Interventions paragraph: the notion of a submodel appears in counterfactuals. Interventional distributions are defined through submodels. This should be brought forward.

  * (C8) Interventions paragraph: The common notation is P(Y | do(X = x)), not P(Y | do(X) = x).

  * (C9) The counterfactual reasoning description is quite cursory (which is perhaps due to considerations of space). However, somewhere, the authors need the emphasize the wildly different nature of quantities that are counterfactuals. The authors give the example of the probability of necessity (PN), while many other more canonical types are usually considered. A good example is the effect of treatment on the treated (ETT). PN is almost never identifiable, while ETT is under lack-of-confounding assumptions.

  * (C10) Typo: Bellman optimally equations -> Bellman optimality equations.

  * (C11) Typo: Rl methods -> RL methods.

  * (C12) Definition of Causal RL -> isn't it also the case that Causal RL sometimes changes the objective of interest? Here I am thinking of counterfactual randomization ideas from [2]. That is, sometimes the ETT-type of quantity may be optimized instead of a total effect.

  * (C13) page 9, paragraph "We note ...": Here again there is a reference to TBOW. A more clear discussion on the difficulties of learning various elements of the causal structure would be desirable. A discussion on various quantities that can be inferred according to Layers 1, 2, and 3 of the causal hierarchy can be found in [1]. Echoing some of these discussion would be rather useful.

  * (C14) page 11: "using clustering outcomes as a causal representation" -> it would be desirable to expand here slightly. From the current writing it is unclear if this is really a causal method or whether it is just a dimensionality-reduction method (there are plenty of non-causal ones).

  * (C15) Table 1: I wonder how useful the last column of the table is, given it is not discussed in the text as I understand. For example, MIMIC-III appears. This is a super-broad hospital / ICU dataset, and I am not sure how useful a pointer to this is, in the current writing. Perhaps references to the table can be added, or the content adapted.

  * (C16) page 12: Abduction-Action-Prediction (AAP) -> throughout the text, counterfactual reasoning is used a single block. The number of different quantities that exist within Layer 3 of the causal hierarchy is vast. For some of them, AAP can only be performed if the true underlying SCM is available. Such requirements are incredibly strong, since SCM noise parameterizations are not unique, and cannot be inferred from any data (see [1] again perhaps). These considerations should be made explicit.

  * (C17) The limitations section does not discuss identifiability issues of counterfactuals. This is one of the main limitations of this approach, and should be stated as such.

  * (C18) For the part on generalizability: An interesting part of closely related literature is that on transportability [3]. Transportability is a graphical approach that encodes exactly which mechanisms of the SCM remain unperturbed in a new environment, and provides rules for who such invariances could be leveraged. Definitely worth mentioning here, I believe.

  * (C19) page 19, Addressing Confounding Bias: Valtorta and Shpitser references prove completeness of do-calculus for _inferring Layer 2 quantities from observational data_. A subsequent paper [4] proves completeness for Layer 3 quanties from obs. data. Finally, recent papers [5, 6] show completeness for Layer 2 / 3 from a combination of observational and experimental data. The latter works seem quite related, since RL may often have observational and experimental data simultaneously.

  * (C20) page 21, last paragraph: this part discussed combining observational and experimental data. I believe this theme could be highlighted as an idea in itself. It features prominently in the causal inference literature, and maybe deserves highlighting in the review as well.

  * (C21) Addressing Selection Bias (SB): There is a good amount of work on addressing selection bias using graphical models [7]. I am surprised this work is not mentioned, since the tools may well be useful for a Causal RL setting.

  * (C22) Does the HER setting really constitute SB? The described phenomenon of changing the reward distribution sounds different than what is usually considered SB in the causal literature.

  * (C23) page 23, last paragraph: TBOW reference appears again for counterfactual reasoning. Furthermore, Kusner and Zhang references are mentioned for fairness, which is fine. However, a number of other results have appeared since, some directly related to fairness in decision-making settings. For example, in the context of causal bandits [8], or in the context of outcome-control [9]. A recent review appears in [10].

  * (C24) page 24, first paragraph, "Through counterfactual reasoning we can explore fairness...": there is a huge amount of complexity hidden here, and the term counterfactual reasoning is used very cursory. A recent organization of various quantities appearing in the literature can be found in [11, Fig. 12].

  * (C25) Section 6.4.1: quotation marks around ' The term "counterfactual" ' need to be fixed.

  * (C26) Section 6.4.1: A note on counterfactual explanations: this name is also used for various approaches that are not at all causal, for example in the algorithmic recourse literature. This is perhaps worth pointing out / being explicit about.

  * (C27) Section 6.4.2: Once again, more recent and closely realted references for fairness+RL can probably be found.

  * (C28) Section 7.1: I personally find that structural learning (inferring the causal diagram) is extremely difficult in practice. I have used structural learning in practice many times, and never obtained anything that is remotely useful in practice. I wonder if it is worth discussing some of the limitations. Structural learning for Semi-Markovian models is very involved. Also, in this context, partially oriented graphs [12] could be discussed/mentioned as a future direction.

   * (C29) Paragraphs 7.5 and 7.6 are quite interesting discussions.


### References

[1] Bareinboim, Elias, et al. "On Pearl’s hierarchy and the foundations of causal inference." Probabilistic and causal inference: the works of Judea Pearl. 2022. 507-556.

[2] Bareinboim, Elias, Andrew Forney, and Judea Pearl. "Bandits with unobserved confounders: A causal approach." Advances in Neural Information Processing Systems 28 (2015).

[3] Pearl, Judea, and Elias Bareinboim. "External validity: From do-calculus to transportability across populations." Probabilistic and causal inference: The works of Judea Pearl. 2022. 451-482.

[4] Shpitser, Ilya, and Judea Pearl. "What counterfactuals can be tested." arXiv preprint arXiv:1206.5294 (2012).

[5] Lee, Sanghack, Juan D. Correa, and Elias Bareinboim. "General identifiability with arbitrary surrogate experiments." Uncertainty in artificial intelligence. PMLR, 2020.

[6] Correa, Juan, Sanghack Lee, and Elias Bareinboim. "Nested counterfactual identification from arbitrary surrogate experiments." Advances in Neural Information Processing Systems 34 (2021): 6856-6867.

[7] Bareinboim, Elias, Jin Tian, and Judea Pearl. "Recovering from selection bias in causal and statistical inference." Probabilistic and Causal Inference: The Works of Judea Pearl. 2022. 433-450.

[8] Huang, Wen, Lu Zhang, and Xintao Wu. "Achieving counterfactual fairness for causal bandit." Proceedings of the AAAI Conference on Artificial Intelligence. Vol. 36. No. 6. 2022.

[9] Plecko, Drago, and Elias Bareinboim. "Causal Fairness for Outcome Control." arXiv preprint arXiv:2306.05066 (2023).

[10] Carey, Alycia N., and Xintao Wu. "The causal fairness field guide: Perspectives from social and formal sciences." Frontiers in Big Data 5 (2022): 892837.

[11] Plecko, Drago, and Elias Bareinboim. "Causal fairness analysis." arXiv preprint arXiv:2207.11385 (2022).

[12] Perkovic, Emilija. "Identifying causal effects in maximally oriented partially directed acyclic graphs." Conference on Uncertainty in Artificial Intelligence. PMLR, 2020.

---

> ### Author Response · Authors · 2023-10-01
> **Response to Reviewer 64gv**
>
> Thank you very much for your thorough review and insightful feedback on our manuscript! We have carefully addressed each of your points and made necessary revisions, as outlined below:
>
> > (C1) Pointers to chapters in Causality.
>
> Thank you for your thoughtful feedback. In the updated manuscript, we have included more references to the Causality book and added pointers to the specific chapters being referenced."
>
> > (C2, 3) Improve the example and discussion on the distinction between RL and Causal RL.
>
> To better demonstrate the intersection of causality and RL, we have replaced the scurvy example with real-world applications more closely related to reinforcement learning (robot manipulation, medical treatment, and autonomous driving). We have included a new figure (figure 1 on page 2) and enhanced the discussion in Section 1 to better analyze the limitations inherent in traditional reinforcement learning (the second paragraph on page 2).
>
> > (C4) Remove the footnote of Markovian assumption.
>
> We have removed the footnote for clarity, retaining only the point that the Markovian assumption is prevalent in human discourse (the Causality book, Chapters 2.2 & 2.9.1).
>
> > (C6) Distributions $P(v_i | pa(v_i))$ are usually called Markov kernels.
>
> We have introduced the term causal Markov kernel [1] in the revised manuscript (the first paragraph on page 5). The term causal mechanism is reserved, as it is commonly used in some recent literature [2].
>
> > (C7) Definition of submodels
>
> Thank you for your thoughtful suggestion! We have incorporated the relevant submodel descriptions into the Intervention section (page 5).
>
> > (C9，16，17)  Different quantities that exist within Layer 3 and their identifiability
>
> In the revised manuscript, we have emphasized that there exist various types of counterfactual quantities within layer 3 (the second paragraph on page 6), and the AAP procedure is not always feasible in practice due to the lack of knowledge (the footnote on page 21). Furthermore, we have highlighted the identifiability challenges associated with counterfactual quantities in the last paragraph of Section 7.1 (page 27).
>
> > (C12) Isn't it also the case that Causal RL sometimes changes the objective of interest?
>
> We agree that RL sometimes changes the objective of interest, e.g., optimizing ETT-type quantity instead of the total effect. This aspect is explicitly emphasized in the revised manuscript for clarity (the third paragraph on page 8). The overall goal of causal RL remains consistent: enabling agents to make more informed and effective decisions.
>
> > (C13) Discussion on the difficulties of learning various elements of the causal structure would be desirable.
>
> Thank you for your suggestions regarding the identifiability issue. In the revised manuscript, we have added an introduction to the Causal Hierarchy Theorem (CHT) and cited the recommended reference to provide a clearer discussion of the difficulties of learning various elements of the causal structure (the last paragraph on page 8).
>
> > (C14) Discussion on Sontakke et al. (2021)
>
> The approach presented in this paper draws inspiration from the principle of independent causal mechanisms. This method induces experimental behavior aimed at facilitating the discovery of causal representations through clustering performance. Clustering, as a part of it, is not a causal approach. We have added a clearer description in the revised manuscript (the second paragraph on page 20).
>
> > (C15) The last column (Environments or Tasks) of the tables.
>
> This column is primarily intended to help the reader quickly navigate through common experimental environments or datasets in causal RL. We have categorized all the environments and datasets that appear in these tables in Appendix B, along with brief introductions, links, and references. For instance, a reader can quickly learn about MIMIC-III from Appendix B.4 Healthcare and directly access the associated codebase using the provided links (page 50).
>
> > (C18) Transportability worth mentioning
>
> Thanks for bringing up the topic of transportability! This is a very interesting but rarely studied topic in reinforcement learning, closely related to generalization ability. We have added a brief introduction to this topic and the related literature in the last paragraph of Section 3 (page 13).
>
> > (C19, 20) Highlight the idea of combining observational and experimental data
>
> In the revised manuscript, we have highlighted the importance of data fusion. In particular, we have summarized it as a notable trend that a number of causal RL methods have investigated, demonstrating promising initial results (the second paragraph on page 18).

---

> > ### Author Response · Authors · 2023-10-01
> > **Response to Reviewer 64gv (cont.)**
> >
> > > (C21, 23, 24, 27) Reference on selection bias and causal fairness
> >
> > Thank you for your additions regarding selection bias and causal fairness. We have incorporated your suggestions by adding the appropriate references to our paper and providing brief introductions to these references (the fifth paragraph on page 18, the last paragraph on page 24, and the third paragraph on page 25). In particular, while there is a growing body of literature exploring the intersection of causality and fair decision-making, only a few of them employ reinforcement learning techniques. We have highlighted this as a future direction in the revised manuscript.
> >
> > > (C22) Does the HER setting really constitute SB?
> >
> > Selective bias occurs when data samples fail to represent the target population. In HER, the algorithm relabels the goal of each collected trajectory and computes new rewards, thereby allowing the agent to learn from failure experiences —— instances where it failed to reach the original goal but succeeded in reaching the relabeled one. However, the relabeled target distribution may not accurately represent the original target distribution. This discrepancy can misguide agents trained with HER, leading them to mistakenly believe that repeating decisions made in the failure experience will lead to high rewards.
> >
> > > (C26) The name "counterfactual explanations" is also used for various approaches that are not at all causal.
> >
> > We have highlighted this in the revised manuscript (the footnote on page 24).
> >
> > > (C28) partially oriented graphs could be mentioned as a future direction
> >
> > In the revised manuscript, we have included an introduction to partially directed graphs, along with appropriate references. We emphasize that this represents a significant future direction in causal RL (the second paragraph on page 28).
> >
> > > (C5, 8, 10, 11, 25) Typos and edits
> >
> > Thank you for pointing out these issues, we have revised and checked the paper according to your comments.
> >
> > Once again, we would like to express our gratitude for your valuable insights, which have undoubtedly contributed to the improvement of our work.
> >
> > [1] Peters, Jonas, Dominik Janzing, and Bernhard Schölkopf. _Elements of causal inference: foundations and learning algorithms_. The MIT Press, 2017.
> >
> > [2] Schölkopf, Bernhard, et al. "Toward causal representation learning." _Proceedings of the IEEE_ 109.5 (2021): 612-634.

---

### Review · Reviewer_6Lf7 · 2023-09-13

**Summary Of Contributions:**

This paper provides a comprehensive summary of recent developments in an emerging discipline of causa reinforcement learning (CRL). In other words, CRL studies applying causal inference theory to reinforcement learning algorithms, incorporating prior knowledge about the underlying causal mechanisms. This allows the AI systems/agents to address various challenges and biases arising from the underlying data-generating processes, including the presence of unobserved confounding, selection bias, and generalizability of policies to new environments where external validity does not hold. The authors provide a catalog of CRL algorithms addressing these challenges. More specifically, Section 3 discusses methods to improve the sample efficiency of existing RL algorithms by exploiting underlying sparse state representation; Section 4 describes CRL algorithms to generalize the effects of candidate policies to different environments and tasks; Section 5 studies the challenges of spurious correlations to existing RL algorithms, including unobserved confounding and selection bias. Finally, Section 6 discusses related topics beyond policy optimization, including explainability and algorithmic fairness.

**Audience:**

Yes

**Broader Impact Concerns:**

This paper provides a comprehensive survey of existing algorithms and methods exploring the intersection between causal inference and reinforcement learning. No new algorithm is proposed. Its longterm societal impact is unclear to see.

**Claims And Evidence:**

Yes

**Requested Changes:**

- [C1] Add simple examples demonstrating the challenges of unobserved confounding, selection bias, and generalizability to new environments.
- [C2] Organize algorithms in each section according to their problem settings, such as off-policy, online, offline-to-online (transfer), and imitation learning.
- [C3] Further clarify the meaning of "sample efficiency" in Section 3. Is it referring to online or offline learning? If it is offline, how does it differ from the discussion in Sections 4 and 5?
- [C4] Generalizability of policies to new environments has been studied in the causal inference literature under the rubrics of transportability. Several conditions and algorithmic procedures have been introduced. The authors should include a few paragraphs discussing related works. See (Bareinboim & Pearl, PNAS-16) for a detailed survey.

Minor comments:
- [C5] The authors could be more gentle when introducing the causal Markov factorization in Eq. (1). It relies on Markovianity. It does not necessarily hold when unobserved confounders exist in the underlying causal model. I recommend starting with a general factorization, including the exogenous parents $U_i$, and then deriving the Markov factorization when Markovianity holds.

**Strengths And Weaknesses:**

This paper attempts to provide a catalog of existing algorithms in the field of causal reinforcement learning. This is quite an undertaking since CRL is undergoing rapid development, and a common solution pattern is still emerging. The authors managed to identify some key challenges facing standard RL, which could be addressed using causal inference techniques. A comprehensive list of related work is provided.

Still, there are some questions and confusion regarding the authors' proposed categorization. Some RL researchers and practitioners might find this paper a bit abstract and confusing. The presentation and organization of the manuscript could be improved. More specifically, I summarize my concerns below.

- The need to adopt causal inference techniques in reinforcement learning tasks could be better motivated. It is unclear how existing RL algorithms are unable to address the challenges described in this paper and how causal inference allows one to overcome these challenges. I would recommend the authors starting by describing basic models (e.g., MDPs) and algorithms for RL and providing simple examples, one for each, demonstrating how these RL algorithms do not lead to satisfactory performance when some critical assumptions are violated, e.g.,  the presence of unobserved confounding, selection bias, and generalizability to new environments.

- The categorization within each section could be confusing. For instance, the "Category" column in Table 2 seems a bit arbitrary, and some concepts are unclear. What does the "Mixed" category mean here, and how does it differ from "Dynamics" and "Tasks"? Also, there are overlaps between the second row, "MDP: OPE" and the third-row, "POMDP." The backdoor/frontdoor adjustment could be seen as a form of off-policy policy evaluation by adjustment. How do they differ, for instance, from the bounding approach in (Namkoong et al., 2020)? Overall, this paper could be most improved by consistently categorizing algorithms across sections. I recommend organizing algorithms according to their problem settings, e.g., off-policy (offline) learning, online learning, offline-to-online (transfer) learning, and imitation learning. The authors could then further categorize algorithms according to their techniques and assumptions.

- Discussions in Section 3 could be improved. It is unclear what type of sample efficiency the authors are referring to. Is it about minimizing the regret in online learning? Or is it about "zero-shot generalization ability and require only a small number of training samples to converge in new environments?" As for the latter case, how does it differ from Section 4? Are some of these discussions in Section 4 redundant? There exist methods exploiting the sparsity of the underlying causal relationships to accelerate online learning processes. Overall, I found those algorithms similar to algorithms for factored MDPs in the RL literature. The ability to exploit the underlying independence in the interventional distributions is not unique to causal inference but also exists in other disciplines, such as hierarchical RL and influence diagrams. Due to these reasons, I suggest moving Section 3 to later parts of the manuscript, e.g., Section 6.

---

> ### Author Response · Authors · 2023-10-01
> **Response to Reviewer 6Lf7**
>
> Thank you very much for your thorough review and insightful feedback on our manuscript! We have carefully addressed each of your points and made necessary revisions, as outlined below:
>
> > (C1) Add simple examples demonstrating the motivations.
>
> To better illustrate the motivation of causal RL, we have replaced the scurvy example with real-world applications more closely related to reinforcement learning (robot manipulation, medical treatment, and autonomous driving). We have included a new figure (figure 1 on page 2) and enhanced the discussion in Section 1 to better analyze the limitations inherent in traditional reinforcement learning (the second paragraph on page 2).
>
> > (C2) Organize algorithms according to their problem settings.
>
> Thanks for the thoughtful advice! In the revised manuscript, we have labeled the problem setting of each paper in Tables 1-4, including offline, online, offline-to-online, and imitation learning. In cases where an approach employs online training augmented by offline datasets, it is labeled as online$^*$.  We have also changed the category "Mixed" in Table 1 into "Dynamics and Tasks" for improved clarity.
>
> As for the difference between backdoor/frontdoor adjustment and the bounding approach in (Namkoong et al. 2020), the key is whether the causal effect is identifiable. When the causal effect is identifiable, we can employ methods such as backdoor/frontdoor adjustment. In cases where the causal effect is unidentifiable, we can resort to set identification (partial identification), deriving its upper and lower bounds, which allows us to assess the robustness of our estimates against unobserved confounding.
>
> > (C3) Moving Section 3 to later parts of the manuscript.
>
> In the revised manuscript, Section 3 has been relocated to Section 5. We have highlighted that the literature discussed in this section is not limited to minimizing regret, but also includes papers related to knowledge transfer (the second paragraph on page 19). The ability to reuse knowledge contributes to both generalization and sample efficiency. For papers that study generalization, sample efficiency improvements on target domains often appear as a byproduct. In our literature review, we emphasize the different motivations and present the methods with a targeted discussion to avoid meaningless repetitions in the two sections.
>
> We agree that part of the work presented in this section is closely related to Factored MDPs. Indeed, if the edges of the dynamic Bayesian networks implied by Factored MDPs represent causal relationships,  then these MDPs essentially serve as causal models. In addition, we also agree that the ability to exploit the underlying independence in the interventional distributions is not unique to causal inference. Causal inference provides a language for formally describing the problems of interest, but it is not the only way to solve these problems.
>
> > (C4) Discussion of transportability.
>
> Thanks for bringing up the topic of transportability! This is a very interesting but rarely studied topic in reinforcement learning, closely related to generalization ability. We have added a brief introduction to this topic and the related literature in the last paragraph of Section 3 (page 13).
>
> > (C5) A gentle introduction of the causal Markov factorization.
>
> Thank you for your kind advice! In the revised manuscript, we have provided a gentler introduction to the causal Markov factorization following your suggestion (the last paragraph on page 4).
>
> We would like to express our gratitude once again for your valuable insights, which have unquestionably contributed to the improvement of our research.

---

### Review · Reviewer_H8ZM · 2023-09-13

**Summary Of Contributions:**

This paper is entitled "Causal Reinforcement Learning: A Survey". It introduces some background concepts on causality and RL, presents a broad definition of the umbrella term causal RL, and presents an overview of existing works grouped by the challenges they try to address: sample efficiency, transfer, spurious correlation, and other considerations (explainability, fairness, safety). The paper concludes with open problems and future directions for the field.

The paper's main contribution is:
 - a broad survey of existing works
 - establishing connections via a problem-oriented taxonomy

**Audience:**

Yes

**Claims And Evidence:**

No

**Requested Changes:**

Critical:

 - rework the narrative, in particular the introduction, and present a more nuanced view of causal RL with a less assertive tone
 - fix the technical ambiguities and inconsistencies
 - make it clearer what are the motivations (hopes, beliefs) for causal RL, what has been achieved and demonstrated so far (successes), what are the costs and limitations of causal RL, and what remains to be done

Strengthen the work:

 - improve the structure of the paper
 - add missing works to the survey

**Strengths And Weaknesses:**

### Strengths

 - the survey compiles a wide range of existing works in causal RL, which is a significant contribution in itself
 - the problem-oriented categorization of these works also makes sense and I believe can be useful to newcomers and practitioners

### Weaknesses

 - the introduction (Section 1) is problematic. I find it is very poor at motivating what causality can bring to the field of RL, and is in fact quite misleading. It does not even mention the requirements and restrictions of most causal RL approaches (prior causal knowledge), but rather presents all of the beliefs of the field ("causality can significantly improve the robustness and interpretability of machine learning models") as indisputable facts. Such a one-sided view is ok for an opinion paper, but it is particularly problematic for a survey paper aimed at newcomers. Instead of providing a fair overview of the promise, challenges and open questions, it perpetuates the delusion that causality is a magic tool that will solve all of RL's problems, which is in the end detrimental to the field. See my detailed comments.
 - the background (Section 2) contains technical ambiguities and inconsistencies (definition of causal graph, initial state as exogenous variable). It proposes the MDP setting as a unified framework to discuss causal RL approaches, which makes little sense as this setting alleviates most of the difficulties causal RL approaches try to tackle, such as confounding or transfer to new tasks (see my detailed comments).
 - the sections "Why causality helps [...]" are too assertive, in the sense that they state "causality helps" (indisputable fact) "here is why", without any nuance (see my comments on Section 3.2). Again, presenting such strong assertions without nuance about the limitations in a survey paper is I think harmful.
 - the structure of the paper is redundant and confusing at times (e.g., 6.1.1 and 6.4.1, see detailed comments)
 - the content of the section "Open Problems and Future Directions" does not fit the description, as it mostly enumerates further existing works that do not fit the proposed taxonomy. I'd suggest to re-work that section and better sort what are actual open problems and future directions. Even better, a section named "Limitations and future directions" would make more sense, as in general all of the presented problems (sample efficiency, transfer, spurious correlation, explainability, fairness, safety) are still open problems.

I have also noted some relevant recent works on causal RL that are missing from the survey:
 - Gun et al. ICML 2022 Provably Efficient Offline Reinforcement Learning for Partially Observable Markov Decision Processes
 - Pace et al. ICML 2023 Delphic Offline Reinforcement Learning under Nonidentifiable Hidden Confounding
 - Gasse et al. TMLR 2023 Using Confounded Data in Latent Model-Based Reinforcement Learning

### Detailed comments

 - p1 "emerge" -> typo

 - p2: "causality can significantly improve the robustness and interpretability of machine learning models" -> I find this whole paragraph rather ambiguous and not well motivated. The difference between "pure data-driven" methods, "causality" and "causal inference" is not clear. In the proposed example with scurvy, a classic online RL algorithm such as REINFORCE or Q-learning does solve the problem optimally, without the explicit need for causality. The problem here relies on the data at hand, experimental data versus observational data. Causality offers a framework / a set of tools to express this problem, but not necessarily a way to solve it. Based on purely observational data, causal inference is in general impossible. I find that this paragraph does a very poor job at explaining the essence of the problem, and rather lures readers into the delusion that causality is a generic tool that can just be applied on top of any ML model, and will improve their robustness as by magic. This is especially problematic for a survey paper targeted at newcomers in the field.

 - p2 "in most RL studies, agents are only allowed to intervene in action variables" -> why is this a problem? For which purpose is this a problem? The general goal in RL is to find an optimal policy for an agent that can act only using the action variables. What would be the benefit of knowing what would happen if one could act on other variables, if in the end these variables are not actionable? It seems like the narrative here is "RL needs causality because RL can not learn causality". What is missing here is, why would a full causal model benefit RL in the first place? Again, I find this paragraph unconvincing and very poor at motivating what causality can bring to existing RL algorithms.

 - p2 "In contrast, with causal RL, you would begin by analyzing the causal relationships and making informed assumptions" -> Same comment as before. This example is misleading. Traditional (online) RL requires no assumptions and leads to an optimal policy, given enough data is collected. Causal RL as described here requires either causal assumptions, which can be wrong (superstition, wrong beliefs), or causal knowledge, which in turn requires experimental data to acquire. Causal reasoning on top of wrong assumptions can be harmful. This is a key limitation to all causal RL methods, and deserves to be part of the discussion. Again, I find this narrative quite confusing, very one-sided, and misleading towards the belief that causal inference always brings benefits to RL.

 - p3 "the essence, implications, and opportunities of causal RL" -> what about the limitations?

 - p4 "definitionn" -> typo

 - p4 "when the parent set PA(Vi) is too small" -> I don't see how small parent sets might violate a Markovian property

 - p4 "the omitted variables influence multiple variables in V" -> what is an omitted variable? Is it the same as an exogenous variable?

 - p4 "Markovian property" -> has not been defined yet. Which Markovian property?

 - p4 "latent variables" -> are latent variables equivalent to exogenous variables in your SCM formulation? I find it confusing to refer to exogenous, omitted and latent variables without clarification.

 - p5 Figure 3a -> I find this representation of probability distributions hard to read. How do these pie charts represent probability distributions?

 - p5 "sailors would be" -> would have been

 - p8 figure 4 -> In the POMDP setting the policy does not have access to the full state, and can not be represented as $do(a \sim \pi(\dot | s_t))$. POMDPs require a different formalism for the policy. As a suggestion, it would be simpler to present only the POMDP regime, which is more general, and derive the MDP regime as a special case (with constraint $O_t=S_t$). Also, one of the main problems that causality tries to address is confounding, which does not happen in the MDP setting. One more reason to focus on POMDPs.

 - p8 "initial state can be considered an exogenous variable" -> I do not follow this reasoning. How can the state, reward and action be endogenous variables, and $S_0$ be an exogenous variable at the same time? Is $S_0$ observed or not to decide on the initial action $A_0$? Is this the MDP or the POMDP setting?

 - p9 "On-policy RL directly learns the causal effects of actions on the outcome from interventional data." -> Then is on-policy RL already a causal RL method? Is causality only useful in the offline setting?

 - p9 "Fortunately [...] we can identify the desired causal effect through certain causal reasoning techniques" -> this is not always true (identifiability problem). This sentence is misleading.

 - p10 "Why Causality Helps Improve Sample Efficiency" -> The paper's title suggests a survey paper, but this section really sounds like an opinion piece. The general tone is 'causality improves sample-efficiency, without question, and here is why'. This is a strong statement, which I find is put forward without due caution. I understand the belief expressed in this section, but I find it problematic that it is presented here as a general truth. The usefulness of causal approaches for sample-efficiency, and for RL in general, yet remains to be demonstrated in the general setting, both in theory and in practice. I strongly suggest changing the tone in this section, and add more nuance, e.g., "How can causality improve sample efficiency?"

 - p10 "causality helps reduce the complexity" -> This is the motivation for a family of approaches, not a general truth. It should be presented as such. "most / many causal RL approaches are motivated by the idea that exploiting the true causal structure of the problem will reduce the complexity of learning, and thus improve sample efficiency"

 - p10 "non-causal methods often demonstrate lower efficiency compared to causal RL methods" -> Again, this assertion is not a universal truth. I'd suggest a less assertive tone: "in some controlled settings, non-causal methods have been shown to demonstrate lower efficiency compared to causal RL methods", or "(Seitzer et al., 2021) show that methods exploiting causal influence for exploration improve sample-efficiency in robotic manipulation tasks"

 - p10 Section 3.3 -> I like this section, which gives a comprehensible summary of existing works. The sub-sections however are highly redundant to the discussion in Section 3.2. I would suggest to merge Sections 3.2 and 3.3 for readability, or simply remove Section 3.2 all together.

 - p22 Section 6 "Considerations Beyond Return" -> I find the structure of this section confusing. 6.1.1 "Leveraging Causality for Explainability" is redundant with 6.4.1 "Explainable Reinforcement Learning with Causality", then 6.2.2 "Leveraging Causality for Fairness" is redundant with 6.4.2 "Fair Reinforcement Learning with Causality", same with 6.3.2 and 6.4.3 about Safety. I suggest to merge these sections.

 - p27 Section 7 -> This section is named "Open Problems and Future Directions", but in reality it is a continuation of the survey of existing works. How is "7.2 Causality-aware Multitask and Meta Reinforcement Learning" an open problem, while "4.3.1 Generalize to Different Environments" or "4.3.2 Generalize to Different Tasks" are not? This structure does not maker much sense to me.

 - p31 "causal RL is a promising method" -> Causal RL is not a single method, as stated in definition 2.3.

---

> ### Author Response · Authors · 2023-10-01
> **Response to Reviewer H8ZM**
>
> Thank you very much for your thorough review and insightful feedback on our manuscript! We have carefully addressed each of your points and made necessary revisions, as outlined below:
>
> > (C2, 3, 4) Improve the introduction's clarity on the role and limitations of causality in RL.
>
> To better illustrate the motivation of causal RL and the intersection of causality and RL, we have replaced the scurvy example with real-world applications more closely related to reinforcement learning (robot manipulation, medical treatment, and autonomous driving). We have included a new figure (figure 1 on page 2) and enhanced the discussion in Section 1 to better analyze the limitations inherent in traditional reinforcement learning (the second paragraph on page 2). In the revised manuscript, we have also highlighted the consequences of wrong assumptions as one of the issues that causal RL needs to investigate, and added additional discussion in Section 7.1 (the second paragraph on page 27).
>
> > (C7) How small parent sets might violate a Markovian property?
>
> Here we are quoting Judea Pearl's discussion of the Markov assumption in [1], Chapter 2.2:
>
> "If a set $\text{PA}_i$ in a model is too narrow, there will be disturbance terms that influence several variables simultaneously and the Markov property will be lost."
>
> That is to say, exogenous variables affecting more than one endogenous variable exist in this case. Therefore, $V_i$ may not remain independent of its non-descendant variables when conditioned on $\mbox{\textbf{PA}}(V_i)$.
>
> > (C8, 10) Relationship among omitted variables, exogenous variables and latent variables.
>
> In the revised manuscript, we have clarified the connection between these three terms (the second paragraph on page 4). Omitted variables is an informal term that denotes factors that are excluded from the system for abstraction in this context. SCM refers to these factors as exogenous variables, which influence the values of the endogenous variables. If there are exogenous variables affecting more than one endogenous variable, the model loses the Markov property. If we explicitly express such variables as latent variables in the causal graph, the Markov property can be restored.
>
> > (C9) Definition of Markovian property.
>
> In the context of SCM, this implies that $V_i$ is independent of its non-descendant nodes when conditioned on its parents $\mbox{\textbf{PA}}(V_i)$.
>
> > (C11, 13) Improve Figure 3a and Figure 4.
>
> We have enhanced the caption for Figure 3a (page5) in the revised manuscript and modified Figure 4 to focus on POMDPs (page 8).
>
> > (C14) Why the initial state can be considered an exogenous variable?
>
> Thank you for pointing this out. We have removed this confusing statement.
>
> > (C15) Is on-policy RL already a causal RL method?
>
> On-policy RL can sometimes learn the total effects of actions on the outcome from interventional data, but it does not directly constitute a causal RL approach. It is also incorrect that causal RL is only useful in the offline setting. Even in the online setting, there are still many issues that require knowledge about causality, such as how to collect intervention data more efficiently, how to transfer acquired knowledge, and how to analyze discrimination that occurs in a decision-making process. Lots of the research presented in Sections 3 to 6 is conducted in the online setting, as indicated in Tables 1-4.
>
> > (C17, 18, 19) Adjust the assertive tone of the sample efficiency section.
>
> We have revised the section names and texts you mentioned to provide a more nuanced perspective that includes discussions on limitations. Thank you very much for your valuable opinions!
>
> > (C20, 21) Merge redundant sections for clarity
>
> We have merged the sections according to your comments for readability.
>
> > (C22) Revise Section 7 to better reflect the content.
>
> In the revised manuscript, we have changed the name of Section 7 to "Limitations and future directions", as you suggested, and added Section 7.1 (page 27), dedicated to outlining the limitations of current methods. In addition, we have explicitly emphasized that the problems discussed in Sections 3-6 remain open for further investigation (the last paragraph on page 26).
>
> > (C24) Additional relevant works on addressing confounding bias
>
> We have incorporated discussions on the first two references (Guo et al., 2022 and Pace et al., 2023 ) you pointed out (the third paragraph on page 16). The third reference (accepted by TMLR on Aug, 2023) is an updated version of a paper we have already cited. Therefore, we have updated the reference to reflect this new version.
>
> > (C1, 5, 6, 12, 16, 23) Typos and edits.
>
> Thank you for pointing out these issues, we have revised and checked the paper according to your comments.
>
> Once again, we would like to express our gratitude for your valuable insights, which have undoubtedly contributed to the improvement of our work.
>
> [1] Pearl, Judea. _Causality_. Cambridge university press, 2009.

---

### Author Response · Authors · 2023-10-01
**Summary of Changes in the New Version**

We sincerely thank each reviewer for your time and constructive comments! Your insightful comments provide valuable guidance for improving the clarity and quality of our work. Causal RL is undergoing rapid development (Reviewer 6Lf7) and we have put a substantial amount of effort into writing this survey (Reviewer 64gv). We appreciate that reviewers acknowledge that our work is informed (all reviewers), well-written with good summarization of existing works (Reviewer nERm and 64gv), provides intuitive examples (Reviewer nERm), and has high empirical value to inspire future RL works (Reviewer nERm and H8ZM).

Here is a summary of the major changes we have made for your convenience:
- We have improved section 1 by introducing examples more closely related to RL and a discussion of the limitations of traditional RL methods.
- We have condensed Section 2.1 by keeping only the core concepts and discussing other concepts in the context of RL in later sections. To be self-contained, a supplementary introduction to causality is included in the Appendix.
- The original section 3 is moved to later parts of the manuscript (Section 5) for clarity.
- We have removed redundancies in Section 5 and Section 6.
- We have renamed Section 7 as "Limitations and Future Directions" and added Section 7.1 dedicated to the limitations of the causal RL methods investigated in Sections 3 to 6.
- We have expanded our discussion to include topics like transportability, data fusion, and the identifiability of counterfactual quantities. We have also incorporated the references suggested by the reviewers, including a number of new papers published in the last few months.

Thank you once again for your valuable feedback, and please let us know if you have any further comments or suggestions.

---

### Decision · Action_Editor_X2ov · 2023-11-16

**Recommendation:** Accept as is

**Comment:**

This paper provided comprehensive list of related work on the causal reinforcement learning (Reviewer 6Lf7, 64gv), and promising research directions (64gv). The current version of the submission has been improved based on previous comments (H8ZM), and reviewers remaining concerns  have been addressed by the authors (nERm).

Although there is still a concern about the framing and categorization of the causal RL, as a survey, I agree with most of the reviewers that there is no solid reason for rejection.

I suggest to accept the paper as is.

**Audience:**

This survey is a good starting point for the researchers who are interested in causality in RL.

**Claims And Evidence:**

Most of reviewers agree that the raised concerns have been addressed.